# Computational Design and Preliminary Serological Analysis of a Novel Multi-Epitope Vaccine Candidate Against Onchocerciasis and Related Filarial Diseases

**DOI:** 10.3390/pathogens10020099

**Published:** 2021-01-21

**Authors:** Robert Adamu Shey, Stephen Mbigha Ghogomu, Cabirou Mounchili Shintouo, Francis Nongley Nkemngo, Derrick Neba Nebangwa, Kevin Esoh, Ntang Emmaculate Yaah, Muyanui Manka’aFri, Joel Ebai Nguve, Roland Akwelle Ngwese, Ferdinand Ngale Njume, Fru Asa Bertha, Lawrence Ayong, Rose Njemini, Luc Vanhamme, Jacob Souopgui

**Affiliations:** 1Department of Biochemistry and Molecular Biology, Faculty of Science, University of Buea, Buea 99999, Cameroon; sheynce@gmail.com (R.A.S.); stephen.ghogomu@ubuea.cm (S.M.G.); Cabirou.Mounchili.Shintouo@vub.be (C.M.S.); neba.nebangwa@ubuea.cm (D.N.N.); mayaahemma@gmail.com (N.E.Y.); muyanuimanka@gmail.com (M.M.); joelebai@gmail.com (J.E.N.); nhon.akwelle@yahoo.com (R.A.N.); ngalenjume@yahoo.com (F.N.N.); 2Department of Molecular Biology, Institute of Biology and Molecular Medicine, IBMM, Université Libre de Bruxelles, Gosselies Campus, 6040 Gosselies, Belgium; Luc.Vanhamme@ulb.be; 3Frailty in Ageing Research Group, Vrije Universiteit Brussel, Laarbeeklaan 103, B-1090 Brussels, Belgium; Rose.Njemini@vub.be; 4Department of Gerontology, Faculty of Medicine and Pharmacy, Vrije Universiteit Brussel, Laarbeeklaan 103, B-1090 Brussels, Belgium; 5Department of Microbiology and Parasitology, Faculty of Science, University of Buea, Buea 99999, Cameroon; francis.nkemngo@crid-cam.net; 6Centre for Research in Infectious Diseases (CRID), Department of Parasitology and Medical Entomology, Yaounde BP 13591, Cameroon; 7Division of Human Genetics, Health Sciences Campus, Department of Pathology, University of Cape Town, Anzio Rd, Observatory, Cape Town 7925, South Africa; eshkev001@myuct.ac.za; 8Department of Public Health and Hygiene, Faculty of Health Science, University of Buea, Buea 99999, Cameroon; fabetty82@yahoo.com; 9Malaria Research Unit, Centre Pasteur Cameroon, Yaoundé Rue 2005, Cameroon; ayong@pasteur-yaounde.org

**Keywords:** *Onchocerca volvulus*, Ov-DKR-2, chimeric antigen, IgG, vaccine development

## Abstract

Onchocerciasis is a skin and eye disease that exerts a heavy socio-economic burden, particularly in sub-Saharan Africa, a region which harbours greater than 96% of either infected or at-risk populations. The elimination plan for the disease is currently challenged by many factors including amongst others; the potential emergence of resistance to the main chemotherapeutic agent, ivermectin (IVM). Novel tools, including preventative and therapeutic vaccines, could provide additional impetus to the disease elimination tool portfolio. Several observations in both humans and animals have provided evidence for the development of both natural and artificial acquired immunity. In this study, immuno-informatics tools were applied to design a filarial-conserved multi-epitope subunit vaccine candidate, (designated Ov-DKR-2) consisting of B-and T-lymphocyte epitopes of eight immunogenic antigens previously assessed in pre-clinical studies. The high-percentage conservation of the selected proteins and epitopes predicted in related nematode parasitic species hints that the generated chimera may be instrumental for cross-protection. Bioinformatics analyses were employed for the prediction, refinement, and validation of the 3D structure of the Ov-DKR-2 chimera. In-silico immune simulation projected significantly high levels of IgG1, T-helper, T-cytotoxic cells, INF-γ, and IL-2 responses. Preliminary immunological analyses revealed that the multi-epitope vaccine candidate reacted with antibodies in sera from both onchocerciasis-infected individuals, endemic normals as well as loiasis-infected persons but not with the control sera from European individuals. These results support the premise for further characterisation of the engineered protein as a vaccine candidate for onchocerciasis.

## 1. Introduction

Onchocerciasis, also called river blindness, remains one of the greatest debilitating and stigmatizing yet neglected tropical diseases. The etiologic agent, *Onchocerca volvulus* (a parasitic filarial nematode) is known to cause principally severe skin and ocular manifestations, including irreversible unilateral or bilateral blindness/visual loss and varying degrees of skin disease [1]. The infective larval stages of the parasite are transmitted through the repeated bites of infective black flies of the genus *Simulium*. These later give rise to adult worms that dwell in subcutaneous tissues in the human hosts where they can survive for up to 15 years (even under drug pressure), with adult female worms hatching approximately 1600 microfilariae daily [2]. Though infections were previously associated mainly with skin and eye lesions, recent records show a trend towards increased mortality. Results from data collected over a 27-year period reported a 5.9% mortality risk attributable to onchocerciasis with greater excess mortality associated with a microfilarial load in younger hosts [3]. Also, other studies have reported that blindness in adults led to a significant increase in mortality and reduced life expectancy [4]. A growing body of evidence has also suggested an association between onchocerciasis and cases of epilepsy [5,6,7,8,9]. With approximately 20.9 million people currently infected, including 14.6 million people with Onchocerca skin disease (OSD) and 1.15 million with vision loss, onchocerciasis remains a major public health problem and a key constraint to socio-economic development especially in Africa [10,11,12].

The creation of large-scale control programmes implementing various strategies targeting both vector and parasite has led to the elimination of the disease in four of the six previously-endemic countries in the Americas and in secluded foci in Africa [13,14]. Though the initial target date to stop treatment in all foci in all six endemic countries in the Americas was 2012, onchocerciasis; however, remains prevalent in Venezuela and Brazil despite the fact that the Onchocerciasis Elimination Programme for the Americas (OEPA) implemented a biannual and tetra-annual distribution of ivermectin (IVM) in contrast to the annual treatments in Africa where the endemic foci are relatively larger and even inaccessible in some cases [15,16]. On the other hand, onchocerciasis elimination in Africa faces, amongst other challenges, the possible emergence of parasite resistance to ivermectin which is the only effective drug currently used for control [17], the contraindication of ivermectin in patients co-infected with high titres of *Loa loa* [18], the emergence of recrudescence [10,19], the absence of robust diagnostic tools [20], the lack of safe and efficacious vaccines [10], and the potential emergence of resistance to the antibiotic doxycycline which targets the *Wolbachia* endosymbiont of the parasite [21]. The situation in Africa may therefore be precarious considering that greater than 99% of the infected and over 96% of the 198 million people at risk live in Africa [22]. In addition, the current COVID-19 pandemic has led to the suspension of neglected tropical disease (NTD) control/elimination programmes by the World Health Organization (WHO), and this will potentially impact the current drive towards onchocerciasis elimination [23,24]. Though the WHO initially set an ambitious target for elimination of onchocerciasis in Africa by 2025, the numerous challenges faced by the disease elimination programme in Africa has led to the conclusion that onchocerciasis will not be eliminated in Africa using the current control tools [25,26]. Besides, mathematical models predict that it may not be possible to achieve onchocerciasis elimination even after 50 years of annual ivermectin treatments in Africa, depending on drug treatment compliance and levels of parasite transmission in endemic areas, [27,28]. This will require billions of ivermectin treatments and will cost millions of US dollars [29,30]. It has been suggested that it is a necessity to complement current chemotherapeutic programs with vaccination in order to reach the set elimination goals [31]. In line with this suggestion, modelling studies have forecasted that a vaccine would protect the substantial investments made by present-day and past onchocerciasis control programmes, decreasing the chance of disease recrudescence (which is already being reported) and offering an important additional tool to mitigate the potentially devastating impact of emerging ivermectin resistance—which is another major concern hindrance to mass drug administration (MDA) programmes [32]. In line with the development of a suitable vaccine, a new Trans-Atlantic partnership, the Onchocerciasis Vaccine for Africa (TOVA) Initiative, was established to develop and test an onchocerciasis vaccine for Africa [31]. Lustigman et al. [10] reported a down-selection process of previously characterised antigens by TOVA that led to the selection of two protein antigens, Ov-103 and Ov-RAL-2 for further clinical development based on proven efficacies of these two in animal model studies. Currently, the goal of TOVA is the production and testing of a river blindness vaccine through Phase I clinical trials by 2022 and Phase 2 efficacy trials by 2030 [31,33]. Though plans are underway to test the efficacies of Ov-RAL-2 and Ov-103, disappointing results often obtained during human proof-of-concept clinical trials continue to highlight the challenges/limitations of making reliable predictions about how well vaccine candidates translate successfully from animal models to humans. More research towards the design and development of more efficacious vaccines is therefore still mandatory. Multi-epitope vaccines (comprising B and T cell epitopes) have been proposed as a way out to evade the shortcomings of whole-organism and single-antigen subunit vaccines [34]. Benefits have been reported for vaccines based on B and T cell epitopes, and these include amongst others, induction of a specific immune response while simultaneously averting the side effects of other unfavourable epitopes in the complete antigen, improved safety and more potent immuno-protection [35,36,37]. Research studies towards to the development of a vaccine for onchocerciasis have targeted mainly the microfilariae (mf) and the infective L3 larval stages, with several studies assessing either irradiated whole-parasites [38,39,40] or single recombinant antigens [41,42]. However, the immunological responses generated in most of these studies have been inadequate to be harnessed in the development of an effective onchocerciasis vaccine [39]. In addition to a prophylactic vaccine targeting the infective L3 larva stage to prevent establishment of disease, a therapeutic vaccine targeting the mf stage will be of importance in limiting the pathological manifestations and also reducing transmission of the disease [43]. Multi-epitope vaccine candidates which are a novel approach to generate more potent immunity have been designed and tested for other nematode infections including lymphatic filariasis and trichinellosis with significant efficacies [34,44,45,46]. However, at the moment, no study has reported the immunological characterisation of any epitope-based chimeric vaccine for onchocerciasis.

Therefore, this work responds to numerous calls to develop novel tools to achieve the current onchocerciasis elimination goal, as outlined in the WHO Roadmap for NTD Elimination. Several studies in humans and animal models have reported that immunity to both *Onchocerca volvulus* mf and L3 is associated to B and T cell responses, with the involvement of antibody-dependent cellular cytotoxicity (ADCC) mechanisms [47,48,49,50]. This work focuses on the computational design and preliminary immunological characterisation of a novel multi-epitope subunit vaccine candidate for onchocerciasis—generated from eight lead *Onchocerca volvulus* protective larval proteins of proven immunogenicity; previously scored based on a system described by Lustigman et al. [10]. The approach is shown in Figure 1. One of the lead protective proteins, Ov-CPI-2 was eliminated because of the presence of its homologue in humans (with which it shares about 29% identity), which may lead to autoimmune responses [51]. Ov-CPI-2 was therefore not used for epitope prediction. These eight proteins have been tested in *Onchocerca*-mouse and *Brugia*-gerbil models, and two of them, namely Ov-RAL-2 and Ov-103, have been selected for future studies in humans [10,51]. In addition, another protective protein Ov-ALT-2 whose homologue in *Brugia malayi* was reported elsewhere to induce high levels of protection was also analysed [52,53]. The selected proteins or their related homologues have been reported to exert anti-L3 and/or anti-microfilaria activities in in-vitro assays and, therefore, could be good targets for prophylactic/therapeutic vaccine development [51]. The design and development of a multi-epitope vaccine with antigenic determinants from the selected proteins could deliver the requisite ‘final punch’ to eliminate human infection with *O. volvulus* completely, as suggested by Nutman [54].

## 2. Results

### 2.1. Protein Sequences Retrieval and Preliminary Analyses

Following the elimination of Ov-CPI-2, due to its potential to generate autoimmune responses, the amino acid sequences of eight selected lead candidate vaccine proteins (Ov-103, Ov-RAL-2, Ov-ASP-1, Ov-ALT-1, and Ov-ALT-2, Ov-B20, Ov-RBP-1 and Ov-CHI-1) were retrieved from the UniProt database, analysed and used to design Ov-DKR-2, a novel multi-epitope vaccine candidate against onchocerciasis. The functional sequences used for epitope prediction were obtained from the selected candidates following signal peptide cleavage as predicted by SignalP 5.0. The conservation between related nematode species was evaluated based on the calculated percentage of amino acid identity (Table 1). The functional sequences of the proteins were then assessed for the presence of B and T cell epitopes on different servers. For multi-epitope vaccines, since the traditional carriers and adjuvants are associated with poor efficacy, vaccine designs with built-in adjuvants have been proposed. Therefore, a built-in adjuvant exhibiting both the functions of a transmission system and a traditional adjuvant was constructed within the vaccine to improve the immunogenicity of epitope peptides by stimulating the innate immune response required for an adaptive immune response [55]. The *Mycobacterium tuberculosis* 50S ribosomal protein L7/L12 (RL7_MYCTU) P9WHE3 retrieved from the UniProt database for use as a built-in adjuvant on the basis of the fact that it is a potent TLR-4 agonist [56] and protective immunity to the larval stages of *Onchocerca volvulus* has been reported to be dependent on Toll-like receptor 4 [41]. (Figure 2).

### 2.2. Linear B-Lymphocyte Epitope Prediction

Linear B-cell epitopes to be used for the chimera design were predicted using the BCPREDS server. In total, 12 high-scoring B-cell epitopes, each, 20 amino acids long were predicted from the primary input sequences and used in the chimera vaccine candidate (Table 2). By selecting the highest-scoring linear B-cell epitopes, the potential of the chimeric vaccine candidate to enhance humoral immunity as well as cell-mediated immunity is assured [57].

### 2.3. Helper T Lymphocytes (HTL) Epitope Prediction

High binding MHC-II epitopes predicted on the NetMHCII 2.3 server for human alleles (HLA-DR, HLA-DQ and HLA-DP) based on their IC_50_ scores were defined as helper T lymphocyte (HTL) epitopes. Twelve high-binding HTL epitopes were selected to be used for the chimeric vaccine candidate (Table 2). Some sequences in the predicted B-cell epitopes had sequences that overlapped with HTL and cytotoxic T lymphocyte (CTL) epitopes. Epitopes having overlapping sequences and as such were fused to form epitopes with contiguous amino acids. No high-affinity epitopes were predicted for Ov-ALT-1, Ov-ALT-2 and Ov-B20.

### 2.4. Cytotoxic T Lymphocytes (CTL) Epitope Prediction

An overall of 89 CTL (9-mer) epitopes were predicted from the selected antigens on the NetCTL 1.2 server at the default threshold score for epitope identification. Nineteen top-scoring predicted CTL epitopes were utilised to generate the final multi-epitope vaccine candidate. It was noted that the antigen, *Ov*-ALT-1 did not possess any CTL epitopes at the default threshold set on the NetCTL 1.2 server.

### 2.5. Conservation of Selected Proteins and Epitopes in Related Nematodes

A BLAST query of the selected proteins on the UniProtKB reference proteomes and Swiss-Prot database showed a high degree of conservation of the selected proteins in selected nematode species (*O. ochengi, O. flexuosa, L. loa, B. malayi* and *W. bancrofti*). BLAST analyses showed that the conservation level for all the selected proteins was generally high across the selected nematodes (ranging from 36.0–100%). However, the homologues for two proteins, Ov-B20 and Ov-CHI-1 were missing for *Onchocerca flexuosa* in the UniProt database. Higher levels of conservation were found in *Onchocerca ochengi,* one of the parasite species that causes bovine onchocerciasis (Table 1). Multiple sequence alignment of the predicted B- and T-lymphocyte epitopes also revealed a great degree of conservation within the selected homologues. The conservation of epitopes (amino-acid percentage identities) ranged from 0% to 89.0%. The epitopes and their percentage identities (in brackets) are presented in Table 2.

### 2.6. Multi-Epitope Chimeric Subunit Vaccine Construction

The linear B-epitopes and T-epitopes predicted from the various constituent proteins were fused using GPGPG and GGGS linkers which have been reported in previous studies to aid in antigen processing and presentation [58]. The TLR4 (PDB ID: 4G8A) agonist 50S ribosomal L7/L12 (Locus RL7_MYCTU) (accession number P9WHE3) was selected as an adjuvant and added to the N-terminus of the designed vaccine candidate using an EAAAK linker to improve on the magnitude of the elicited antigen-specific immune responses [59]. Meanwhile, an 8x-His tag was added at the C-terminal of the designed protein to aid in protein purification (through immobilized-metal affinity chromatography) and identification (through Western blot analyses). The final vaccine candidate generated was composed of 821 amino acid residues (Figure 2). This multi-epitope is designated as ‘Ov-DKR-2’, and this name was used in this paper to refer to the designed chimeric vaccine antigen.

### 2.7. Physiochemical Properties Prediction

The final chimeric protein had a predicted molecular weight (MW) of 87.3 kDa with a theoretical isoelectric point value (pI) of 8.52. As deduced from the pI, the protein is thus slightly basic in nature. The half-life was predicted to be 30 hours in mammalian reticulocyte extracts in vitro, more than 20 hours in yeast and more than 10 hours in *E. coli*, in vivo. The predicted instability index (II) was 33.16, classifying the protein as stable. The estimated aliphatic index was predicted to be 63.59—signifying thermostability; since higher aliphatic index values are reported to be related to the thermostability of a protein [60]. The predicted Grand average of hydropathicity (GRAVY) was −0.551. The negative GRAVY value indicates the protein is hydrophilic in nature and has the potential to interact with water molecules [61].

### 2.8. IFN-γ Inducing Epitope Prediction

For the adjuvant, a total of 122 potential IFN-γ-inducing epitopes (overlapping 15-mer sequences) were predicted, although none of these scored above the default threshold set for epitope prediction. However, for the main vaccine construct, a total of 668 potential epitopes were generated, having both negative and positive prediction scores using the IFNepitope server, which predicts epitopes using the MERCI software. From the main vaccine construct a total of 86 IFN-γ-inducing epitopes with positive scores between 1 and 3 were predicted. The predicted presence of IFN-γ inducing epitopes was consistent with the simulated IFN-γ levels produced after immunization with the vaccine candidate using the C-ImmSim server (http://150.146.2.1/C-IMMSIM/index.php) for immune simulation (Figure 3).

### 2.9. Antigenicity and Allergenicity Prediction of Chimeric Vaccine Construct

The VaxiJen 2.0 server predicted an antigenicity score of 0.5568 with a parasite model at a threshold of 0.5 while the ANTIGENpro predicted an antigenicity score of 0.860185. The results from both servers indicate that the generated constructs (with and without adjuvant) are both antigenic in nature. Meanwhile, the generated vaccine candidate (consisting the various epitopes and linkers with/without built-in adjuvant) was predicted to be non-allergenic on both AllerTOP v.2 and AllergenFP servers. This reduces the potential development of allergenic effects during clinical trials.

### 2.10. Secondary Structure Prediction

Secondary structure analyses predicted the designed vaccine candidate to contain 28% alpha helix, 7% beta strand and 64% coil (Figure 4A). Concerning the solvent accessibility of amino acid residues, 56% were exposed, 19% partially exposed while 23% were predicted to be buried. The large proportion of exposed amino acid residues indicates the potential of protein in terms of epitopic properties towards possible T cell and B cell activation. The RaptorX Property server predicted that a total of 294 residues (35%) were present in disordered regions. The pictorial presentation of the PSIPRED prediction of secondary structure and the disorder of the final vaccine candidate is shown in Figure 4.

### 2.11. Prediction of Intrinsic Disordered Regions

The protein was predicted on the IUPred server to contain intrinsically disordered regions (IDRs), spanning its entire length. No IDRs were predicted within the adjuvant domain. The presence of IDRs in the protein could further contribute to its antigenicity. The disordered profile for the chimeric protein is displayed in Figure 4B.

### 2.12. Tertiary Structure Modelling, Refinement and Validation

Five tertiary structure models of the protein construct were predicted on the trRosetta server (Figure 5A). The selected model for further refinement and validation was model one, which had a TM of 0.100. This was then refined initially on 3Drefine producing five models and model five (with 3Drefine score: 45297.3; GDT-TS: 0.9951; GDT-HA: 0.9415) was chosen for further refinement on the GalaxyRefine server. The GDT-TS (global distance test-total score) and GDT-HA (global distance test-high accuracy) are global measures of the agreement between a predicted model and the experimental structure [62]. GalaxyRefine server yielded five models. Model three from the GalaxyRefine server refinement analyses was found to be the best based on various parameters including GDT-HA (0.9571), root mean square deviation (RMSD) (0.566), MolProbity (2.391). The clash score was 29.7, and poor rotamers score was 1.4. This model was taken as the final vaccine model for further analysis (Figure 5B). The overall quality factor of the modelled protein was 75.4% using ERRAT. (Figure 5C). Ramachandran plot analysis to validate the refined 3D model of the protein revealed that 96.5% of residues are located in most favoured regions, 2.8% in allowed regions and only 0.7% in the disallowed region (Figure 5D). The ProSA-web server gave a Z-score of ^−^8.88 for the input vaccine protein model. This score lies just outside the score range commonly obtained for native proteins of comparable size-resolved by X-ray crystallography (Figure 5E).

### 2.13. Conformational B-Cell Epitopes Prediction

In total, 448 residues were predicted to be found in 10 discontinuous B cell epitopes with scores ranging from 0.512 to 0.799 on the ElliPro server. The conformational epitopes ranged from 5 to 146 residues, as shown in Table 3.

### 2.14. Molecular Docking of Subunit Vaccine with Immune Receptor (TLR4)

The protein binding and hydrophobic interaction sites on the protein surface were predicted with the aid of the CASTp server (Figure 6). The large binding pocket identified between residues 41–797 could act as a potential binding site for TLR4. The pocket molecular surface area was 2929.0 Å^2^ with a molecular surface volume of 6512.0 Å^3^; the mouth molecular area was about 197.8 Å^2,^ and the molecular circumference sum was 80 Å (Figure 6A). Protein vaccine-mediated targeted docking against TLR4 performed on the FRODOCK server showed an interaction between the protein and its receptor (Figure 6B).

### 2.15. Immune Simulation

Immune simulation performed on the C-ImmSimm server revealed a marked increase in the generated secondary responses. This pattern is, in principle, consistent with the generation of an actual immune response. (Figure 7A). The primary response simulated was marked by high levels of IgM. The secondary and tertiary responses simulated, on the other hand, depicted marked increases in B-cell populations, and levels of IgG1 + IgG2, IgM and IgG + IgM antibodies with a corresponding decrease in the antigen concentration (Figure 7A,B). This indicates the development of immunological memory—evident in the elevated memory B-cell population (Figure 7C)—and isotype switching, which resulted in rapid antigen clearance upon subsequent exposures to the chimeric antigen. A likewise higher response was predicted for the T_H_ (helper) and T_C_ (cytotoxic) cell populations with corresponding memory development upon subsequent exposures to the antigen (Figure 7D–F). Moreover, macrophage, dendritic cell, and natural killer cell populations were elicited and sustained at high levels throughout the immunization period (Figure 7G–I) Repeated exposure, with 12 injections (given at week intervals) showed an increasing level of IgG1, declining IgM levels while IFN- γ levels and T_H_ cell populations were maintained at high levels throughout the course of the exposure. Therefore, Ov-DKR-2 appeared to elicit both humoral and cellular immune responses which have been reported to be vital in protecting against onchocerciasis. Besides, the low Simpson D index (Figure 3) that indicated high diversity in the T-cell response confirms the presence and effectiveness of multi-epitopes.

### 2.16. Codon Optimization of Final Vaccine Construct and Cloning

The Java Codon Adaptation Tool (JCat) was employed for codon optimization in order to ensure maximal protein expression in *E. coli* (strain K12). The codon-optimized sequence which was 2463 nucleotides in length had a codon Adaptation Index (CAI) of 1.0 and the average GC content of 52.7% suggesting the likelihood of good expression for the vaccine protein in the chosen *E. coli* host since the optimal percentage range of GC content should lie between 30% to 70% [63].

### 2.17. Mass Expression and Purification of Recombinant Chimeric Antigens

With the goal of experimentally assessing the in-silico predictions, i.e., the potential of the designed chimeric protein as a vaccine candidate, protein expression was attempted in two *Escherichia coli* strains BL 21 DE3 and SHuffle T7 Express, which are reported to have better expression host for Cys-rich proteins. Preliminary analyses showed better expression profiles in BL 21 DE3 *E. coli,* and the produced proteins were purified using Ni^++^ columns (Figure not shown). The resulting proteins were contaminated by some impurities; the second round of purification using size-exclusion chromatography led to an improvement in protein purity.

### 2.18. Purified Proteins Reacted with Serum from Onchocerca-Exposed, Loiasis and Mansonellosis Patients but not with Non-Endemic European Controls

To assess the potential vaccinogenic and cross-protective properties of Ov-DKR-2, Western blot analyses of the recombinant vaccine were performed using serum pools from cohorts of individuals with different clinical status with regards to nematode infections. Results indicated that the purified chimeric protein was immunoreactive with different reactivity profiles for the different cohorts except for the European control serum pool. The western blot analyses also suggested that the sera also reacted with degraded Ov-DKR-2 protein, and this seems to be more intense for the loiasis-infected and onchocerciasis endemic normal populations (Figure 8A). To evaluate the potential of the purified recombinant antigen in the development of a suitable vaccine candidate for onchocerciasis (and related nematode infections), antigen-specific total IgG responses of serum from the different cohorts described above were investigated by ELISA. Results obtained show that serum from onchocerciasis-exposed as well as serum from individuals infected with other related nematodes reacted with Ov-DKR-2—indicating that the protein contained B-epitopes that were recognized by immunoglobulins in the analysed serum samples. These results were very much in conformity with the Western blot analyses reported (Figure 8B).

## 3. Discussion

Neglected tropical diseases (NTDs) are a collection of diverse bacterial, parasitic, viral, and fungal infections that persist in many tropical and sub-tropical developing countries where the poverty burden is particularly high [64]. Currently, the World Health Organization (WHO) identifies 20 major conditions as neglected tropical diseases (NTDs) [65], up from the 17 previously recognised in the early 2000s. The NTDs are currently prevalent in at least 149 countries worldwide, affecting more than two billion people and constituting an important cause of morbidity, disability, and mortality [66]. Though NTDs have not individually been global priorities, the fact that they jointly exert a global disability burden and human suffering equivalent to that jointly caused by tuberculosis, human immunodeficiency virus/acquired immunodeficiency syndrome (HIV/AIDS), and malaria, they definitely warrant a global response [67,68]. Though chemotherapeutic measures for disease control exist for some of these diseases, drug discovery and development for NTDs is still facing many challenges, amongst which stands the fact that these diseases generally do not constitute a priority for the pharmaceutical industry due to low financial returns [69]. These challenges are further compounded by the fact that drugs on their own are not a sustainable approach to disease due to rates of reinfection, risk of drug resistance, and inconsistent maintenance of drug treatment programs. In an opinion piece, Chami and Bundy (2019) argued that the expansion of medicine donation programmes and the development of new medicines are not the primary solutions to sustaining and expanding the growth of neglected tropical disease programmes [70]. It has generally been agreed that to successfully reach the targeted elimination goals for the NTDs, a combined portfolio of diagnostics, drugs, vaccine and vector control agents have to be deployed [67]. However, to date, there is no clinically-approved vaccine for any of the NTDs. Preventative and/or therapeutic NTD vaccines that will complement the other tools already in use are needed as long-term solutions [27,71]. Lustigman et al. (2017) have reiterated the urgent need for a vaccine [2] to ensure achievement of the onchocerciasis elimination set by Expanded Special Project for Elimination of Neglected Tropical Diseases (ESPEN) [72]. Though many onchocerciasis vaccine candidates (including subunit and whole-parasite vaccines) have already been assessed in animal model studies) [10,51,73,74,75], none has, at the moment, been successfully translated for human use. Though whole-organism vaccines hold more promise for efficacy in challenge studies, the focus has of late shifted towards the development of subunit vaccines as they are generally easier to upscale and associated with lower risks [76]. Epitope-based vaccines epitomize a new approach to generate potent and specific immune responses while averting unneeded responses (like immunopathogenic or immune-modulating responses) from epitopes in the complete antigen [77]. Amongst the advantages of epitope-based vaccines are increased safety, the opportunity to rationally engineer the epitopes for increased potency and coverage, and the possibility to focus immune responses on conserved epitopes [78].

This study consequently focused on the computational design and preliminary in-vitro characterisation of a novel multi-epitope chimeric vaccine candidate against onchocerciasis using epitopes predicted from eight proteins previously tested in preclinical development. [75,79,80]. The selection of previously-characterised proteins, validated in in-vitro studies, eliminates the risk of using non-immunogenic antigens in the design. The vaccine designed in this report could display both prophylactic and therapeutic potentials as the antigens selected for epitope prediction are expressed in both the infective L3 larval and microfilariae stages. This might also be the reason why the protein reacted with sera from both infected and putatively immune individuals. The designed vaccine candidate also possesses the potential for cross-protection since the selected proteins and predicted epitopes used in generating the chimeric peptide showed considerable conservation in the related nematodes analysed. Cross-protection following vaccination could be important to consider since, in most endemic settings, the prevalence of one NTD overlaps with at least one other.

The roles of both the humoral and cellular arms of the host immune system in immuno-protection against *Onchocerca volvulus* have been reported in both clinical and animal model studies [40,49]. The vital function of antibody-dependent cell-mediated cytotoxicity (ADCC) [50,81,82] has also been reported and research on the candidates planned for clinical trials, Ov-RAL-2 and Ov103, reported that ADCC is a crucial mechanism for a successful prophylactic vaccine against this infection [83]. The importance of TLRs in mediating interactions between helminthic parasites and the human host immune system has also been described [84] and the TLR4 pathway has been implicated in host immunity to onchocerciasis [38]. The endosymbiont *Wolbachia,* present in many filarial parasites (including *O. volvulus*) possesses a major surface protein which interacts with the innate immune system through TLR2 and TLR4 leading to an inflammatory response characterized by proinflammatory cytokine expression in murine dendritic cells and macrophages [85]. Away from animal model studies, research with human subjects has also reported reduced expression of TLR1, 2, 4, and 9 on B-cells, both at the mRNA and protein levels have also been reported in human subjects infected with lymphatic filariasis compared to uninfected controls—suggesting that TLR1, 2, 4, and 9 down-regulation might be a vital immune evasion mechanism by filarial parasites during infection [85].

In this study, B and T cell epitopes from selected immunogenic proteins previously tested in animal models were predicted and fused using appropriate linkers to create a novel multi-epitope chimeric antigen. Though previous experiments in animal models have reported the vaccinogenic potential of Ov-CPI-2 [86] and a genetically modified form its *Brugia malayi* homologue, Bm-CPI-2M [87], this protein was eliminated from the epitope prediction analyses because the presence of its homologue in humans (with 29% identity) which could lead to the induction of autoimmune responses. Specialised spacer sequences have been reported to be vital in the design of epitope-based vaccines [35]. With this in mind, previously reported GGGS, GPGPG and EAAAK linkers [60,61,88,89] were suitably inserted between the predicted epitopes to produce sequences with minimized junctional immunogenicity while also ensuring optimal expression and enhanced bioactivity of the chimeric vaccine candidate upon expression in a suitable host. The built-in adjuvant was specifically selected to interact with TLR4 based on the role of this receptor in onchocerciasis immunity.

The designed protein was subject to a series of analyses to further assess its suitability as a potential vaccine candidate. Computational analyses of the designed chimeric vaccine candidate divulged the presence of a large number of high-scoring MHC Class I and B-cell linear epitopes, and high-affinity MHC Class II epitopes. In addition, the final vaccine polypeptide was predicted to contain interferon-γ (IFN-γ)-inducing epitopes. IFN-γ induction has been reported to be an essential mechanism to control *O. volvulus* infection and results in individuals with no dermal microfilariae [90]. In a bid to further evaluate the vaccinogenic potential of the chimera developed through our work, its antigenicity was compared with that of Ov-RAL-2 and Ov-103, which are lead vaccine candidates for onchocerciasis [91]. Ov-DKR-2, the multi-epitope peptide generated in this study showed antigenic scores higher than those of Ov-RAL-2 and Ov-103 both on the Vaxijen v2.0 and ANTIGENpro servers. Strikingly, the generated chimeric protein we generated gave an antigenic score almost twice that of Ov-103 on ANTIGENpro (0.93 for the chimeric peptide and 0.59 for Ov-103). It will, therefore be interesting to evaluate Ov-DKR-2 in challenge experiments in parallel with Ov-RAL-2 and Ov-103. It has been reported that vaccines composed of multiple epitopes are often poorly immunogenic and need to be delivered with adjuvants [35]; however, the designed protein showed similar antigenicity scores in the presence or absence of the built-in adjuvant sequence. The non-allergenic nature of the designed multi-epitope chimera further supports its potential as a vaccine candidate. Finally, the stability profile of the multi-epitope chimera, Ov-DKR-2 was predicted to be higher than those of Ov-RAL-2 and Ov-103. Ov-RAL-2 was predicted to be unstable, suggesting the need for a continuous search of novel vaccine candidates with better stability profiles. The results from the Western blot analyses (Figure 4B) showed that degradation products of the chimeric target protein could not be recognised by the serum from infected individuals but was recognized by other serum pools. This suggests that the degradation products possess epitopes that interact with antibodies present in the other cohorts but absent in actively infected human serum samples. The degradation may have led to the exposure of epitopes that are specifically recognized by other cohorts but not the pool for the infected subjects. The possibility of the detected lower band being a contaminant is also not totally excluded. This will be further investigated in our subsequent experiments.

Preliminary analyses of physico-chemical properties revealed that Ov-DKR-2 has a molecular weight of 87.2 kDa. The protein is predicted to be slightly basic in nature (with a theoretical pI of 8.52) and stable upon expression. The aliphatic index showed that the protein possesses aliphatic side chains, indicative of potential hydrophobicity. These different parameters predict that the recombinant protein is thermally stable and is thus appropriate for use in endemic areas, which are mostly located in sub-Saharan Africa. Secondary and tertiary structure data on a target protein has also been reported to be important in rational vaccine design [35]. Secondary structure prediction analyses by the RaptorX server revealed that the protein comprised largely of coils (64%) with 35% of residues predicted to be located in disordered regions. The presence of disordered regions was further confirmed using the IUPred server, which showed that the disordered regions spanned throughout our chimeric protein sequence but were lacking in the added adjuvant domain. Natively unfolded protein regions (or intrinsically-disordered regions), as well as alpha-helical coiled coil structures, have been reported to be vital forms of “structural antigens”, since they can fold into their native structure and therefore react with antibodies naturally induced in response to infection [92]. The two-step refinement approach used markedly improved the 3D structure of the vaccine candidate, and validation predictions using the different tools showed desirable properties. The Ramachandran plot analyses reveal that most of the residues are located in the favoured and allowed regions (99.2%), suggesting the satisfactory quality of the final 3D model generated.

Previous research has reported the participation of TLR4 in immune protection mechanisms against *O. volvulus* and other pathogens [38,85,93]. We also previously showed that the TLR4 agonist embedded as a built-in adjuvant in a chimeric peptide is able to interact with the receptor TLR4 [94]. The possible immune interactions between TLR4 and the chimeric vaccine peptide were assessed through a protein-protein docking analysis since a TLR4 agonist was embedded as a built-in adjuvant within the designed chimera.

For the in-silico simulation studies, a constant number of 1000 antigens were given per injection, and only the chimeric protein (containing the various epitopes, built-in adjuvant and linkers) was submitted for analyses. Results from the in silico immune simulation analyses showed patterns consistent with typical immune responses. Repeated exposure to the chimeric vaccine candidate, led to an overall increase in the elicited immune responses. In onchocerciasis, IgG_1_, IgG_3_ and IgE responses to parasite antigens have been implicated in immune protection mechanisms [50,86,95]. However, this work assessed only total IgG in immunological assays and the responses to IgG subclasses as well as IgE responses will be assessed in our future experiments. From in silico immune simulation analyses, the development of memory B and T cells was evident and generated memory in B cells persisted for several months. The immune simulation analyses also revealed helper T cells stimulation. Another stimulating observation was the rise in f IFN-γ and IL-2 levels following the first injection, and these remained at peak levels following repeated exposures. This predicts high levels of T_H_ cells and subsequently putatively effectual Ig production, supportive of a humoral response through T-dependent B-cell activation. Although, the built-in adjuvant used was predicted to lack IFN-γ-inducing epitopes which are important for immune protection in onchocerciasis, the mechanism of action of this adjuvant as is yet to be elucidated. Therefore, the adjuvant was maintained on the basis of its ability to interact with TLR-4, which also plays a vital role in protective immune responses. In addition, it is worth noting that the large numbers of the IFN-γ non-inducing epitopes (with negative scores) predicted in the construct indicates the inability of the 15-mers predicted within the designed sequence to induce IFN-γ secretion but not the ability of these sequences to abrogate or inhibit IFN-gamma secretion. The Simpson index D for investigation of clonal specificity points to the generation of a diverse immune response. The diversity of the observed results is reasonable; taking into consideration the fact that the designed chimeric polypeptide comprises of several B and T epitopes. However, the role of CD8+ T cells in host protective immunity to *Onchocerca* microfilariae remains yet unclear, [49]. However, the development of these cells may be a response to stimulation by Th1 and Th2 cells [96] as cytokines secreted from both Th1 and Th2 cells (including IL-2 and IL-4) have been implicated in the development of primary and long-lived memory CD8+ responses [97,98]. Additionally, interferon-γ (IFNγ)-dominant T_H_1-type responses, which have also been reported in putative immune individuals [99,100], are associated with increased numbers of T_H_1 cells, cytotoxic CD8^+^ T cells, neutrophils, and macrophages [101]. Consistently, high CD8 + T cell levels have been reported in putative immune individuals [102]. The results obtained from immune simulation prediction for cytokine proliferation and cellular responses will necessitate validation in immunological assays. However, it is important to indicate that the C-ImmSim server accepts only protein sequences, and it is therefore difficult to make predictions on how the other traditional adjuvants will stimulate the immune system using this server.

One of the crucial steps in validating a vaccine candidate for onchocerciasis is to assess its immunoreactivity through serological analysis, given the important role of antibodies in immune protection against the disease [103]. This step requires that the protein be expressed using recombinant DNA technology in an appropriate host. The *Escherichia coli* expression system, which is easy to use and comparatively cheaper is the preferred choice and common first step for the production of most recombinant proteins [104,105]. In a bid to accomplish optimal expression of Ov-DKR-2 in *E. coli* (strain K12), codon optimization was done. Both the codon adaptability index (1.0) and the GC content (52.7%) following optimization were supportive for high-level expression of the recombinant protein in bacteria. The sequence was further optimised using a GenScript proprietary tool (increasing the GC content to 58%) to further optimise expression in bacteria. Expression was successful in bacteria, and both Western blot analyses and ELISA showed that protein reacted strongly with sera from onchocerciasis-infected patients (HOS) and endemic normal subjects (ENS) as well as serum from low-endemicity regions (IVS and HES). This indicates that the individuals in these two cohorts had antibodies that recognised the epitopes in the protein. It was observed that infected and putatively immune individuals (also called endemic normal, ENS) elicited very strong responses to the target protein, unlike the Ivermectin-treated cohort. Previous studies have reported similar results - with the recognition of many parasite antigens by sera from both infected and putatively immune individuals [106]. Moreover, in vitro studies have reported that sera from infected and putatively immune individuals promoted L3 larval killing and/or limited the moulting of L3 to L4 in the presence of granulocytes [107]. The observed decrease in the reaction of the chimeric protein with the Ivermectin-treated group (IVS) could be due to decreased antigen stimulation following chemotherapy which leads to decreased antibody responses [108]. Similar results were obtained in our previous studies where an increase in compliance with Ivermectin treatment led to reductions in total IgG, IgG subclasses and IgE responses with time [109]. The observed recognition of the chimeric antigen displayed by the HES group suggests possible cross-reaction with other unknown active or past pathologies faced by the Rwandan population as reported in our previous study [109]. The protein was also observed to cross-react with serum samples from the Loa-infected persons (LLS) and a single sample from a Mansonella-infected individual (MPS). This observation suggests that the chimera contains epitopes whose equivalents could be present in homologous proteins in the related nematodes. The ELISA results corroborated the results obtained from Western blot analyses as the fall in the antibody responses due to IVM treatment was reflected in the significantly decreased responses observed in the IVS group compared to the infected subjects (HOS). Both assays showed the low reactivity of serum from the non-endemic European controls (ECS) compared to the other cohorts (Figure 8). In addition, assessment of cross-protection potential showed that the protein contained epitopes recognised by antibodies in the sera of persons infected with loiasis. This is consistent with the conservation of the proteins and the selected epitopes predicted. However, there remains the need to assess the potential cross-protection by the protein in persons infected with other nematodes, including *B. malayi* and *W. bancrofti*. Some extent of degradation was observed for purified protein and the degradation product was also immunoreactive with some of the serum pools of the cohorts used in the study (Figure 8B). The introduction of flexible linkers (including Gly–Ser linkers) in proteins has been reported to be a potential cause of instability which results in degradation [110,111]. The GGGS linker was used in the design of Ov-DKR-2 to connect CTL-epitopes.

## 4. Conclusions

In the first part of this study, a combination of computational tools was employed to design a novel chimeric vaccine candidate containing multiple B cell and T cell (HTL and CTL) epitopes using antigens previously tested in vaccine development efforts and reported to be immunogenic either in vitro or in vivo in animal models. Since both cellular and humoral immune mechanisms have been implicated in responses that correlated with protection against onchocerciasis, the probability of generating an efficient vaccine against onchocerciasis is increased by the design of a vaccine that incorporates epitopes that stimulates both arms of the immune system. Given that the proteins used for epitope prediction in order to design the chimera are expressed in the microfilarial (associated with pathology) and L3 larval (associated with infection) stages of the parasite which are the key targets for parasite vaccine development, the designed vaccine candidate might possibly deliver both prophylactic and therapeutic effects. The chimeric vaccine candidate, therefore, possesses the potential to eventually be used as a complementary tool to achieve onchocerciasis elimination. The potential for cross-protection is also expected (evident from the level of sequence conservation in related nematodes) from the designed vaccine candidate, and this could benefit NTD control programs focusing on related filarial diseases such as lymphatic filariasis and loiasis, as well as control programs for onchocerciasis in livestock. This is supported by our preliminary experimental findings. Additional experiments, chiefly including immunisation and challenge experiments in animal models will be necessary to further assess the vaccinogenic potential of the chimera.

## 5. Materials and Methods

### 5.1. Ethical Clearance and Blood Sample Collection

All blood samples used for this study were either collected under the ethical clearance (N˚2015/01/543/CE/CNRESH/SP) previously obtained or through collaborating partners, and the serum samples are described in our previous publication [109]. The study was done in adherence to the principles of the Helsinki Declaration on the use of humans in biomedical research. Administrative authorization was obtained from the Cameroon Ministry of Public Health (N˚631–1315) and informed consent forms were provided and well explained to all participants who took part in the study. Only the participants willingly signed the consent form prior to participation in the study were allowed to participate in the study. Participation was entirely voluntary and participants were free to withdraw at any time without any coercion. Participants’ confidentiality was respected during data collection, analysis, reporting and reporting.

### 5.2. Protein Selection and Sequence Retrieval for Vaccine Preparation

The full amino acid sequences of the eight different lead *Onchocerca volvulus* protective proteins (*Ov*-103, *Ov*-RAL-2, *Ov*-ASP-1, *Ov*-ALT-1, *Ov*-ALT-2, *Ov*-B20, *Ov*-RBP-1 and *Ov*-CHI-1) were retrieved from UniProt (http://www.uniprot.org/) in FASTA format. These proteins which have all been reported to be immunogenic in animal models were selected based on high scores obtained cognizant of the criteria developed for the selection of antigen for vaccines development [10]. The proteins were then assessed for the presence of signal peptide on the SignalP 5.0 server (http://www.cbs.dtu.dk/services/SignalP/) which discriminates classical secretory and non-secretory proteins. SignalP 5.0 integrates a deep neural network-based approach with conditional random field classification and optimized transfer learning to improve signal sequence prediction across all spheres of life and outperformed other predictors in benchmark tests [112].

### 5.3. Linear B Cell Epitope Prediction

B cell epitopes are antigenic elements that can be recognized and bound by secreted antibodies or B-cell receptors. These epitopes are vital in the design of epitope-based vaccines [113]. The essential role of antibodies in the development of prophylactic immunity against onchocerciasis has been reported [83]. The BCpreds server (http://ailab.ist.psu.edu/bcpred/) [114] was used to predict linear B cell epitopes (20-mers) for the selected proteins. BCPreds server functions by applying a combination of the subsequence kernel (SSK) and a support vector machine (SVM) approach to predict the location of linear B-cell epitopes. BCPreds server projects non-overlapping 20-mer epitopes using 75% specificity via amino acid pairs (AAPs) method with 72.5% prediction accuracy [36]. Only epitopes having a score of 1 were selected for the construction of the final protein.

### 5.4. Prediction of Cytotoxic T Lymphocytes (CTL) Epitopes

Cytotoxic CD8+ T-lymphocytes have been reported to be important in the generation of immuno-protective against *Onchocerca lienalis* microfilariae in a mice model [49]. The NetCTL 1.2 server (www.cbs.dtu.dk/services/NetCTL/) [115] was employed for the prediction of immunogenic CTL epitopes to elicit cell-mediated immunity and form the memory cell pool. The server works on an algorithm that integrates prediction of peptide MHC class I binding, proteasomal C terminal cleavage and TAP transport efficiency. The NetCTL1.2 server can predict CTL epitopes for 12 MHC class I supertypes. The server uses artificial neural networks to predict MHC class I binding and proteasomal cleavage while TAP transport efficiency is predicted using a weight matrix. For this work, only the A1 supertype was chosen for convenience with the default threshold of 0.75 set for epitope identification.

### 5.5. Prediction of Helper T-Lymphocytes (HTL) Epitope

The CD4+ helper T cells perform several roles, from the activation of innate immune system cells, B-lymphocytes, cytotoxic T cells, as well as non-immune cells, and also play key functions in immune reaction suppression [116]. CD4+ helper T-lymphocytes have been reported to be important in the generation of immuno-protective immunity against *Onchocerca species* in both mice and humans [48,49]. HTL epitopes of 15-mer length for human alleles were predicted using the NetMHCII 2.3 Server (http://www.cbs.dtu.dk/services/NetMHCII/) [117]. The NetMHCII 2.3 server uses artificial neuron networks to predict the binding of peptides to human HLA-DR, HLA-DQ, HLA-DP and mouse MHC class II alleles the prediction of MHC II epitopes is based on the affinity for their receptor which can be inferred from the IC_50_ values and percentile ranks assigned to each predicted epitope. The epitopes selected for the construction of the chimera was based on their ability to bind to several MHCII molecules and their low IC_50_ scores.5.6. Conservation of proteins and predicted epitope in related nematodes.

To evaluate the potential of cross-protection by the novel vaccine candidate to selected related nematodes, a BLAST query was done on UniProtKB protein database using the proteins selected and predicted epitopes. Percentage identities for homologues of the selected protein were obtained. The percentage conservation of the different epitopes selected to design the chimera was evaluated using multiple sequence alignment of protein homologues in related nematodes. The nematodes selected for comparison of proteins and epitopes included *Onchocerca ochengi* (which infects cattle), *Onchocerca flexuosa* (which causes onchocerciasis in deer), *Loa loa* (human filarial worm, also known as African eye worm), *Brugia malayi* (one of the three causative agents of lymphatic filariasis in humans), and *Wuchereria bancrofti* (one of the three causative agents of lymphatic filariasis in humans) which were selected on the basis of their importance to human and livestock health.

### 5.6. Construction of Multi-Epitope Vaccine Sequence

Top scoring CTL epitopes, high-affinity HTL epitopes and high-scoring B-epitopes were selected from the epitope prediction analyses and used to design the chimeric vaccine candidate. It was noticed that some epitopes had overlapping sequences that were present in both B cell and T cell epitopes. The different epitopes were linked together through GGGS for CTL epitopes and GPGPG linkers for both HTL and linear-B cell epitopes as previously described. Furthermore, the 50S ribosomal protein L7/L12 (Locus RL7_MYCTU), TLR4 agonist (Accession no. P9WHE3) was chosen as a built-in adjuvant to improve immunogenicity of the designed vaccine candidate and added at N-terminus using the EAAAK linker.

### 5.7. Prediction of IFN-γ Inducing Epitope

Interferons (IFNs) are pleiotropic cytokines with antiviral, antitumor and immunomodulatory properties, being central coordinators of the immune response [118]. The IFNepitope server (http://crdd.osdd.net/raghava/ifnepitope/scan.php) was used to predict 15-mer IFN-γ epitopes for Ov-DKR-2. IFNepitope is an online prediction server that aims to predict and design the peptides from protein sequences having the capacity to induce IFN-gamma release from CD4+ T cells. The server works on a training dataset of 10,433 experimentally validated MHC class II binders or T-helper epitopes from Immune Epitope Database (IEDB which consists of 3705 IFN-γ-inducing and 6728 non-inducing MHC class-II binders, which can activate T-helper cells [119]. Due to limitations in the number of residues that can be used for prediction on the server, IFN- γ epitope predictions were done separately for the adjuvant and the main vaccine construct. The server generates overlapping sequences from the protein inputted, and IFN-γ epitopes are predicted from these with assigned numerical scores. For Ov-DKR-2, the prediction was performed using the motif and support vector machine (SVM) hybrid approach.

### 5.8. Prediction of Vaccine Candidate Antigenicity and Allergenicity

Antigenicity is the ability to specifically combine with the final products of the immune response (i.e., secreted antibodies and/or surface receptors on T cells) [120]. Two different servers. ANTIGENpro and VaxiJen v2.0 were used to predict the antigenicity of the novel chimeric vaccine antigen. ANTIGENpro (http://scratch.proteomics.ics.uci.edu/) works with protein antigenicity microarray data to predict protein antigenicity and generates an antigenicity index. The server accuracy on a combined dataset (of antigens that elicit a strong antibody response in protected individuals but not in unprotected individuals, using human immunoglobulin reactivity data obtained from protein microarray analyses; and known protective antigens from the literature) was evaluated at 76% by cross-validation experiments [121]. To gain more insight into the antigenicity of the multi-epitope vaccine candidate, the VaxiJen 2.0 server (http://www.ddg-pharmfac.net/vaxijen/VaxiJen/VaxiJen.html) which is an alignment-free approach for antigen prediction, based on auto cross-covariance (ACC) transformation of protein sequences into uniform vectors of principal amino acid properties was used. Prediction of antigenicity with the VaxiJen v2.0 server is alignment-free like ANTIGENpro, and the prediction is based on various physiochemical properties of protein [122].

AllerTOP v2.0 and AllergenFP servers were used to predict the potential generation of allergenic responses by the multi-epitope vaccine, The AllerTOP v2.0 (http://www.ddg-pharmfac.net/AllerTOP) server uses amino acid E-descriptors, auto- and cross-covariance transformation, and the k nearest neighbours (kNN) machine learning methods to classify allergens. The method reports a 85.3% accuracy using the kNN method at 5-fold cross-validation [123]. AllergenFP (http://ddg-pharmfac.net/AllergenFP/) on the other hand is an alignment-free, descriptor-based fingerprint approach that uses amino acid E-descriptors and auto- and cross-covariance (ACC) transformation of protein sequences into uniform equal-length vectors to identify allergens and non-allergens. The approach has been applied to a set of 2427 known allergens and 2427 non-allergens and correctly identified 88% of them with a Matthews correlation coefficient of 0.759 [124].

### 5.9. Physiochemical Properties Prediction

The online webserver, ProtParam (http://web.expasy.org/protparam/) [125], was used to predict the physiochemical properties (including molecular weight, amino acid composition, theoretical isoelectric pH, instability index, in vitro and in vivo half-life, aliphatic index, and grand average of hydropathicity (GRAVY)) of Ov-DKR-2.

### 5.10. Secondary Structure Prediction

Two online webservers were used to analyse the secondary structure of the designed multi-epitope subunit vaccine candidate. The PSIPRED 4.0 server (bioinf.cs.ucl.ac.uk/psipred/), a popular and front-line protein secondary structure method was initially used. The PSIPRED method utilizes incorporation of two feed-forward neural networks which perform the investigation on result retrieved from PSIBLAST (Position-Specific Iterated BLAST). PSIPRED 4.0 has an average Q3 secondary structure prediction accuracy of 84.2% [126]. Further analyses of the secondary structure properties of the protein construct was done using the RaptorX Property web server (http://raptorx.uchicago.edu/StructurePropertyPred/predict/), which is a template-free approach. The server uses a novel machine learning model called DeepCNF (Deep Convolutional Neural Fields) to predict secondary structure (SS), solvent accessibility (ACC), and disordered regions (DISO) simultaneously [127]. The DeepCNF machine learning model in addition to modelling complex sequence-structure relationship by a deep hierarchical architecture can also model interdependency between adjacent property labels and can obtain ~84% Q3 accuracy for three-state SS, ~72% Q8 accuracy for eight-state SS [128].

### 5.11. Prediction of Intrinsic Disordered Regions

Intrinsic disordered regions (IDRs) in proteins have been described as structural antigens [92] and several leading vaccine candidates in *Plasmodium* are reported to have extensive regions of disorder [129]. Intrinsic disordered regions were predicted for the multi-epitope protein construct using IUPred (http://iupred.enzim.hu/). IUPred predicts intrinsically disordered regions from amino acid sequences by estimating their total pairwise interresidue interaction energy, based on the assumption that IDR sequences do not fold due to their inability to form sufficient stabilizing interresidue interactions [130].

### 5.12. Tertiary Structure Prediction

The trRosetta server (transform-restrained Rosetta) (https://yanglab.nankai.edu.cn/trRosetta/) which is an algorithm for fast and accurate de novo protein structure prediction was used to predict the 3D (functional) structure of the designed multi-epitope vaccine antigen. A guiding principle of the Rosetta algorithm is to attempt to mimic the interplay of local and global interactions in determining protein structure [131]. The trRosetta server builds the 3D structure of the input protein sequence based on direct energy minimizations with a restrained Rosetta. The restraints include inter-residue distance and orientation distributions, predicted by a deep residual neural network. In benchmark tests on Critical Assessment of protein Structure Prediction (CASP)13 and Continuous Automated Model EvaluatiOn (CAMEO) derived sets, trRosetta outperformed all previously described methods [132].

### 5.13. Tertiary Structure Refinement

Refinement of the predicted 3D model obtained for the multi-epitope chimeric vaccine candidate was done using a two-step approach; initially on the 3Drefine server (http://sysbio.rnet.missouri.edu/3Drefine/index.html) and then on the GalaxyRefine server (http://galaxy.seoklab.org/cgi-bin/submit.cgi?type=REFINE). The 3Drefine refinement protocol uses iterative optimization of hydrogen bonding network combined with atomic-level energy minimization on the optimized model using a composite physics and knowledge-based force fields for efficient protein structure refinement [133]. The protocol method has been widely assessed on blind CASP experiments and on large-scale and assorted benchmark datasets and shows steady improvement over the initial structure in both global and local structural quality measures. The GalaxyRefine server works based on a refinement method that has been successfully assessed in community-wide CASP10 experiments. The method initially rebuilds side chains, executes side-chain repacking and subsequently performs overall structure relaxation by molecular dynamics simulation. The GalaxyRefine method exhibited the best performance in improving the local structure quality in the CASP10 assessment. The method does have an improvement of both global and local structure quality on average when used to refine the models generated by state-of-the-art protein structure prediction servers, including I-TASSER and ROSETTA models [134]. The GalaxyRefine server is based on a refinement method that executes short molecular dynamics (MD) relaxations after repeated side-chain repacking perturbations and has been broadly used in both investigational and computational studies [135].

### 5.14. Tertiary Structure Validation

Model validation procedures constitute a critical step in the model building sequence since they detect potential errors in predicted 3D models [136]. Generally, the accuracy of a predicted structural model determines the range of its potential applications. For instance, it would be almost pointless to use a protein structure for structure-based vaccine design if there is uncertainty about the quality of the target protein 3D model [137]. The ProSA-web (https://prosa.services.came.sbg.ac.at/prosa.php) was used to assess the quality of the refined 3D protein structure. ProSA-web affords an easy-to-use interface to the ProSA program and is regularly used in protein tertiary structure validation procedures. ProSA calculates and assigns a global quality score for any specific input structure, and this is displayed on a plot in the context of all known protein structures whose structures have been resolved by nuclear magnetic ressonance (NMR) or X-ray crystallography. Also, any problematic parts of a structure are shown and highlighted in a 3D molecule viewer. If the calculated score falls outside the range characteristic of native proteins, the structure likely contains errors. Its range of application includes error recognition in experimentally determined structures, theoretical models and protein engineering [138]. In order to further validate the refined 3D structure, the ERRAT server (http://services.mbi.ucla.edu/ERRAT/) was used to analyse non-bonded atom-atom interactions compared to reliable high-resolution crystallography structures [139].

### 5.15. Prediction of Conformational B Cell Epitopes

It has been estimated that >90% of B-cell epitopes are discontinuous, i.e., consisting distantly separated segments that are in the pathogen protein sequence and brought into proximity by the folding of the protein [140,141]. Conformational (discontinuous) epitopes in the validated 3D structure were predicted using the ElliPro server (http://tools.iedb.org/ellipro/). The ElliPro methods implements three algorithms that execute approximation of the protein shape as an ellipsoid, calculation of the residue protrusion index (PI) and lastly clustering of neighbouring residues based on their PI values. ElliPro assigns a score, described as PI value to each output epitope averaged over each epitope residue. For instance, an ellipsoid with PI 0.9 value is considered as 90% protein residues are included while the residual 10% residues lie outside of ellipsoids. For each epitope residues, the PI value is assessed based on the centre of mass of residue residing outside the largest possible ellipsoid. When compared with other structure-based epitope prediction methods, ElliPro gave the best performance; giving an AUC value of 0.732, when the most significant prediction was considered for each protein [142].

### 5.16. Protein-Protein Docking of Vaccine Candidate Antigen with TLR4

The evocation of a proper immune response is reliant on the interaction between an antigenic molecule and a specific immune receptor. The CASTp 3.0 server (http://sts.bioe.uic.edu/castp/) was used to predict binding pockets or cavities found on the TLR-4 receptor. The CASTp server is an online resource for locating, outlining and measuring concave surface regions on three-dimensional structures of proteins. These regions include pockets located on the surfaces of protein, while voids are commonly buried in the interior of proteins. CASTp can be applied to study surface features and functional regions of proteins [143]. In CASTp, voids are defined as buried unfilled empty space inside proteins after removing all hetero atoms inaccessible to water molecules (modelled as a spherical probe of 1.4Å) from outside. Pockets, on the other hand are demarcated as concave caverns with constrictions at the opening on the surface regions of proteins. Unlike voids, pockets allow easy access to water probes from the outside. The CASTp sever is excellent in providing identification and measurements of surface accessible binding pockets along with the information of inner inaccessible cavities in a protein molecule [143]. The FRODOCK 2.0 web server (http://frodock.chaconlab.org/) was used to perform protein-protein docking of the multi-epitope vaccine construct with TLR4 (PDB ID: 4G8A) to evaluate the interaction between the ligand and receptor and consequently the development of immune response. The vital function of TLR4 in protective immunity to the larval stages of *O. volvulus* is reported [38]. In the FRODOCK approach, the projection of the interaction terms into 3D grid-based potentials is combined with the efficiency of spherical harmonics approximations to speed-up the search for protein-protein interactions and the binding energy upon complex formation is approximated as a correlation function composed of van der Waals (VDWs), electrostatics and desolvation potential terms. The interaction-energy minima are identified by a new, fast and exhaustive rotational docking search combined with a simple translational scanning. FRODOCK results gotten on standard protein-protein benchmarks reveal its general applicability and robustness. The accuracy is similar to that of existing state-of-the-art initial exhaustive rigid-body docking tools, but achieving superior efficiency [144].

### 5.17. Immune Simulation

The C-ImmSim server (http://150.146.2.1/C-IMMSIM/index.php) was used to predict the immunogenicity and immune response profile of the novel chimeric vaccine candidate as previously reported [94]. C-ImmSim is an agent-based model dynamic simulator that combines position-specific scoring matrix (PSSM) for immune epitope prediction and machine learning techniques to predict immune interactions. It “simultaneously simulates three compartments that represent three separate anatomical regions found in mammals: (i) the bone marrow, where hematopoietic stem cells are simulated and produce new lymphoid and myeloid cells; (ii) the thymus, where naive T cells are selected to avoid auto immunity; and (iii) a tertiary lymphatic organ, such as a lymph node” [145]. Following the TOVA approach with regards to the target product profile (TPP) of a prophylactic vaccine for onchocerciasis, three injections were given at four week-intervals [146]. All simulation parameters were set at default with time steps set at 1, 84, and 168 (each time step is 8 h and time step one is injection at time = 0). Therefore, three injections were given four weeks apart. Furthermore, to simulate repeated exposure to the antigen seen in a typical endemic area so as to probe for clonal selection, 12 injections of the designed chimeric vaccine candidate were also administered four weeks apart. The Simpson index, D which is a measure of diversity was interpreted from the plot.

### 5.18. In Silico Codon Optimisation, Gene Synthesis and Cloning of Vaccine Candidate Construct

Reverse translation and codon optimization were done on the Java Codon Adaptation Tool (JCat) server (http://www.prodoric.de/JCat) to maximise the production of the multi-epitope vaccine candidate in a suitable expression system. The coding for the designed vaccine candidate was codon-optimised for expression in *E. coli* (strain K12) host since codon usage in *E. coli* is different from that of the *O. volvulus* parasite from where the sequence of final vaccine construct is derived. The output generated by the JCat server which includes codon adaptation index (CAI) and percentage GC content can be used to assess the protein expression levels. The CAI provides information on codon usage biases. The perfect CAI score should be 1.0, but a score of 0.8 or greater can be considered as a good score [147]. GC content of a sequence for protein expression should range in between 30–70%, and GC content values outside this range suggest unfavourable effects on translational and transcriptional efficiencies [63]. The optimized gene sequence of the final vaccine construct was further optimised using a proprietary tool from GenScript Piscataway, USA. It was synthesized and cloned by GenScript into pET-30a (+) vector, with *XhoI* and *NdeI* restriction enzyme sites introduced to the N and C-terminals of the sequence, respectively.

### 5.19. Expression and Purification

*E. coli* strains BL21(DE3) pLysS and Shuffle T7 Express transformed with pET30a+-Ov-DKR-2 and pET30a+-Ov-DKR-2 were grown at 37 °C in 5000 mL of Terrific Broth containing 34 µg/mL chloramphenicol and 50 µg/mL kanamycin. Since preliminary analyses showed a better expression profile for BL21(DE3) bacteria strain, this system was used for the upscaling. Induction of protein expression in bacteria was done using 1mM Isopropyl β-D-1-thiogalactopyranoside (IPTG) at room temperature overnight, with shaking at 190 rpm, and the protein was expressed as a C-terminal 8x-His tagged protein. The bacteria cells expressing recombinant protein were centrifuged from culture medium, and lysis was performed by sonication (40% Amplitude, 10 min) on ice. Due to complications in obtaining the protein in a soluble state, the expressed antigen was purified from inclusion bodies. Following sonication, solubilization of the bacteria pellet was performed in urea as described previously [148]. Briefly, following several washes with 50 mM Tris buffer containing 500 mM NaCl, 2% cholic acid and 2 M urea, solubilisation was achieved by incubating the bacteria pellet in buffer containing 2 M urea at pH 12.5 at room temperature overnight. The urea-solubilised antigen was then purified on Ni++-immobilized-metal affinity chromatographic (IMAC) columns (GE Healthcare) following standard procedures for the purification of His-tagged proteins. To remove contaminating proteins, a further round of purification for pooled Ni++-IMAC purified fractions was achieved by size-exclusion chromatography using the HiLoad 16/600 Superdex 200 pg column on an AktaPurifier (GE Healthcare Life Sciences) following column calibration with protein standards containing Thyroglobulin, MW 670 kDa; Ovalbumin, MW 45 kDa; gamma-globulin, MW 93 kDa and Vit B12, MW 1.35 kDa. The purified chimeric protein then was resolved by SDS-PAGE to assess its purity and integrity while Western blot using monoclonal anti-6xHis antibodies (SigmaAldrich, St. Louis, Missouri, United States) was done to confirm antigen identity. The purified protein fraction concentrations were obtained using the BCA protein assay kit (Pierce, Rockford, IL, United States). For Western blot analyses, 2 μg of protein samples were run on a 4%–12% Tris-glycine polyacrylamide gradient gel (Bio-Rad, Carlsbad, CA) and then transferred to Hybond-C Extra nitrocellulose membranes (GE Healthcare). The membrane was blocked with 5% milk in TBS-NP40 overnight at 4 ˚C followed by incubation with mouse monoclonal anti-6x-His antibodies (1:10,000). After three changes of wash buffer (TBS + 0.005% NP40) done at 5-minute intervals each, the membranes were incubated with rabbit anti-mouse alkaline phosphatase-conjugated secondary antibodies (1:5000) (Promega) for 90 min. Detection was done with nitro blue tetrazolium/5-bromo-4-chloro-3-indolyl phosphate (NBT/BCIP) substrate, and the membrane was monitored for colour development.

### 5.20. Serological Characterization of Ov-DKR-2

Two assay types were employed for the sero-characterisation of the purified antigen. For Western blot, sera pools from persons of different clinical status were created and used at a dilution of 1:800 in 5% milk. Anti-human antibody-ALP conjugate was used as secondary antibody and revelation in both cases was done using 5-bromo-4-chloro-3-indolyl-phosphate in conjunction with nitro blue tetrazolium (BCIP/NBT). To further assess the vaccinogenic potential of Ov-DKR-2, individual serum samples from six subject groups consisting of onchocerciasis-infected individuals and different controls collected from various populations with defined disease status—namely: OVS (human onchocerciasis sera) for individuals currently suffering from onchocerciasis, ENS (endemic non-symptomatic sera) for people who have lived in endemic areas for extended periods without developing any symptoms of the disease, ITS (Ivermectin treatment sera) for samples with at least three years of IVM treatment from the Bandjoun Health District earlier mentioned, LLS (*human loiasis sera*) for individuals infected with actively-infected with *Loa loa* parasite, HES (African hypo-endemic sera) for a population from the Huye District in Rwanda where onchocerciasis is hypo-endemic, and ECS (European control sera) for non-exposed disease-free persons of European origin) were screened for the presence of antigen-specific antibodies. For cohorts used, the numbers were as follows: HOS (27), ENS (21), HES (16), IVS (16), LLS (16) and ECS (3). Checkerboard titration experiments were initially performed to determine optimal antigen/antibody concentrations. Nunc MaxiSorp™ 96-well microtiter plates (Nunc, Roskilde, Denmark) were coated with 2 μg/mL (100 ng) of the purified antigen overnight at 4 °C in 100 mM carbonate buffer, pH 9.6. Plates were washed with three changes of wash buffer (PBS + 0.05% Tween 20), with 5 min between each wash and blocked with 3% non-fat dried milk for 1 h 30 min at 37 °C. Plates were again washed as previously described and incubated with the various serum samples (as a primary antibody) at a dilution of 1:250 for 1 h 15 min at room temperature. Goat anti-human IgG (Fc-specific) peroxidase conjugate (Merck Millipore, Billerica, MA, USA) was then incubated as secondary antibody at a dilution of 1:5000 for 1 h at 37 °C after washing. After a final wash, incubation with the chromogenic substrate was performed for 10 min at 37 °C. The reactions were stopped with 3 M HCl, and the optical densities at 450 nm were recorded using the SPECTROstar Nano (BMG LABTECH, Ortenberg, Germany). All washings and antibody dilutions were performed in wash buffer (PBS + 0.05% Tween-20).

### 5.21. Data Analyses

Normality of distributions was assessed using a Shapiro–Wilk test. Comparisons of multiple groups were made using the Kruskal–Wallis test to determine the statistical difference of multiple groups. A *p*-value *<* 0.05 was considered statistically significant. Scatter plots were generated using Graph Pad Prism 7.0 (La Jolla, CA, USA).

## Figures and Tables

**Figure 1 pathogens-10-00099-f001:**
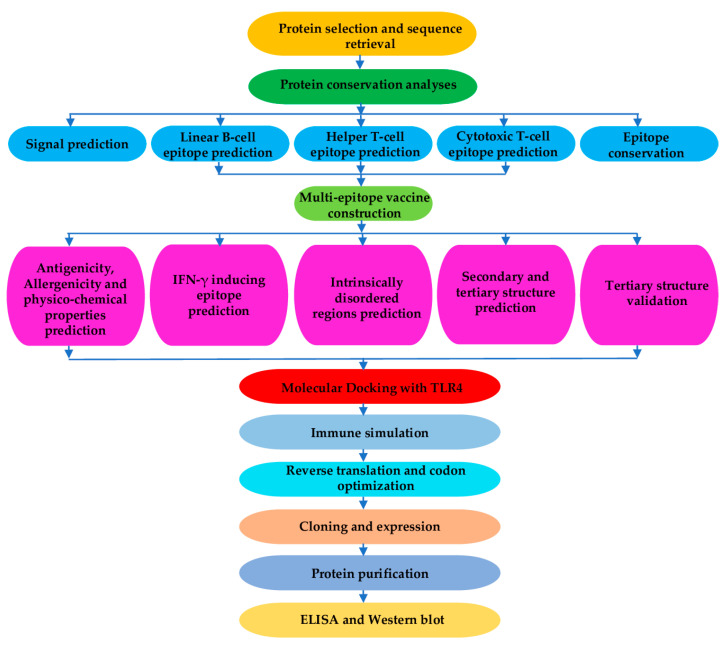
Flowchart for the designed study. The complete approach employed in the study comprises of several phases, which involved retrieving the target protein and performing conservation analysis, epitope predictions (cytotoxic-T lymphocyte (CTL)-, helper-T lymphocyte (HTL)- and linear-B cell epitopes) from the chosen protein; in silico recombinant chimeric vaccine construction, 3D structure modelling, refinement and validation, molecular docking of chimeric antigen with the immune cell receptor TLR4: immune simulation to assess (in silico) the potential of the vaccine candidate to initiate an immune response and the type of immune response elicited. Lastly, codon optimisation, gene synthesis/cloning, protein expression/purification and immunological assays (ELISA and Western blot) were performed to characterize the serological response to the antigen.

**Figure 2 pathogens-10-00099-f002:**
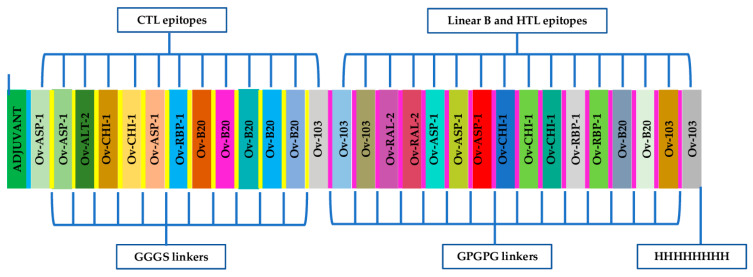
Schematic presentation of the generated multi-epitope chimeric vaccine antigen, Ov-DKR-2. The 821-amino acid long polypeptide sequence comprising a built-in adjuvant (green) at the N-terminal connected with the multi-epitope sequence through an EAAAK linker (cyan). Linear B-cell epitopes and HTL epitopes are linked using GPGPG linkers (yellow) while the CTL epitopes are linked through of GGGS linkers (purple). The 8x-His tag was added at the C-terminal for purification and identification processes.

**Figure 3 pathogens-10-00099-f003:**
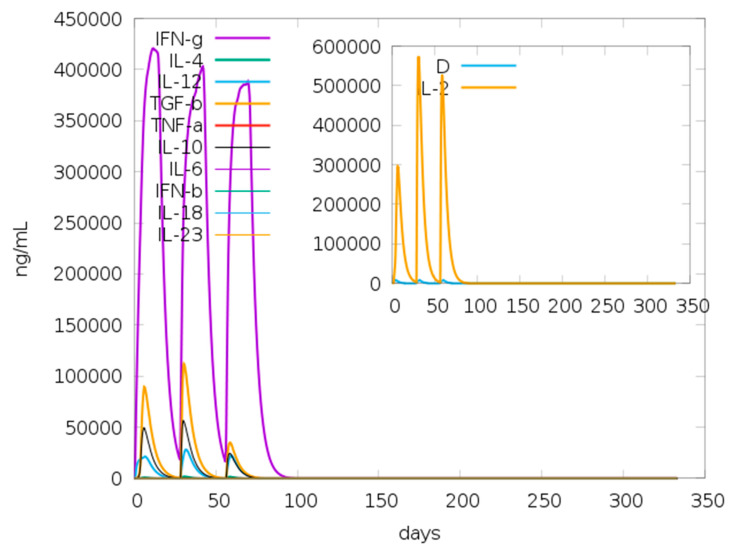
Induced levels of cytokines as simulated with C-ImmSim after three injections, 28 days apart. The main plot shows cytokine levels produced after the injections. The insert plot shows IL-2 level with the Simpson index, D (measure of diversity). The rise in D over time indicates the development of epitope-specific dominant clones of T cells. The bigger the D, the lower the clonal diversity.

**Figure 4 pathogens-10-00099-f004:**
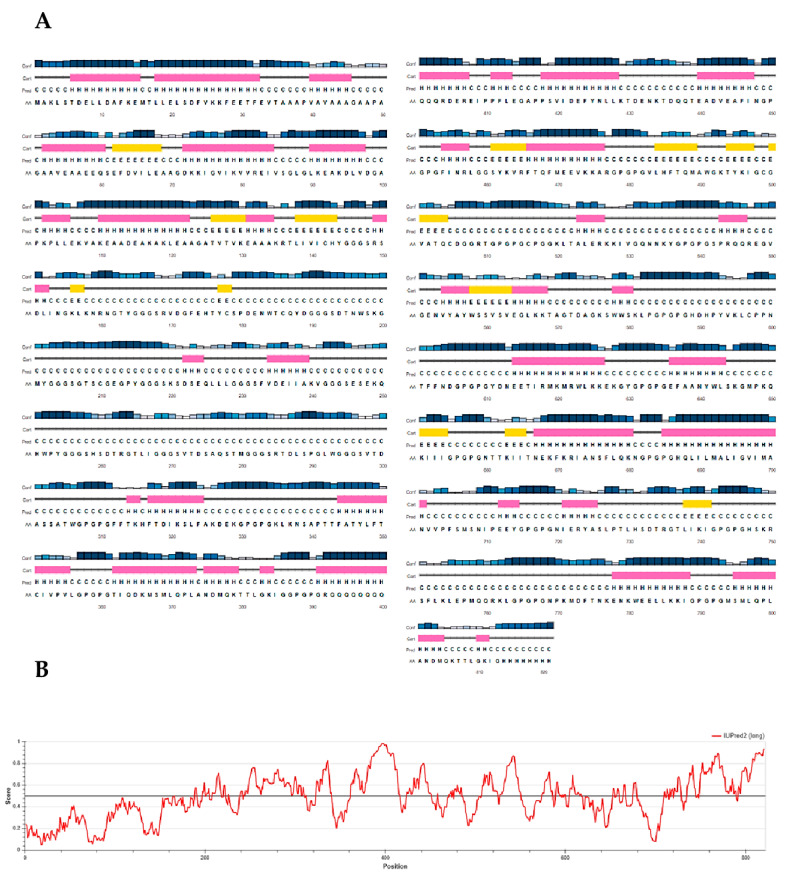
Graphical representation of secondary structure features of the final subunit vaccine sequence. (**A**) The chimeric vaccine antigen is predicted to include 28% alpha-helices, 7% beta strands and 64% coils. The first line from top to bottom (Conf) indicates the level of confidence of prediction; the second line (Cart) indicates the three-state assignment cartoon; the third line (Pred) indicates the three-state prediction and the last line (AA) indicates the target sequence. On the Cart line, yellow stretches denote helices; pink stretches denote strands, and dark lines denote coils. (**B**) 35% of amino-acid residues were predicted to be located in intrinsically disordered regions (IDR) as seen from peaks above the threshold (black line). The IDR span across the protein but are absent in the TLR4 agonist used as an adjuvant (beginning of the graph: aa positions 01 to 130).

**Figure 5 pathogens-10-00099-f005:**
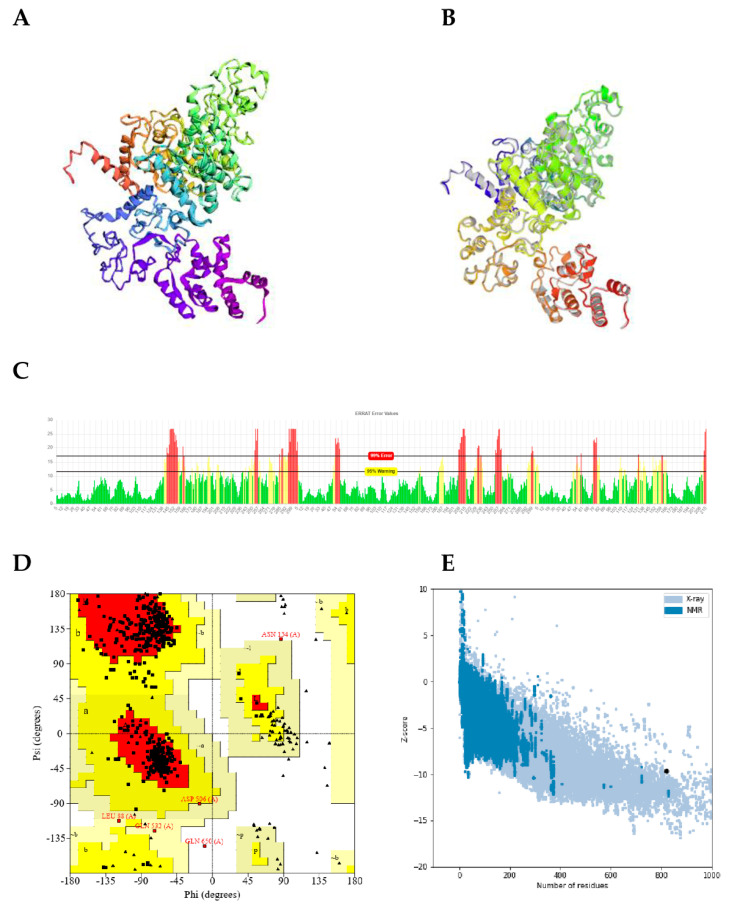
Protein 3D structure modelling, model refinement and validation. (**A**) The 3D model of the multi-epitope vaccine obtained following homology modelling on the trRosetta server. (**B**) Refinement: superimposition of the refined 3D structure (coloured) on the ‘crude model’ (gray) by the GalaxyRefine server. Validation of the refined model showing (**C**) ERRAT plot with a score of 75.3998. (**D**) Ramachandran plot analysis with 96.5%, 2.8% and 0.7% of protein residues in favoured, allowed, and disallowed (outlier) regions respectively and (**E**) ProSA-web, giving a Z-score of ^−^8.88.

**Figure 6 pathogens-10-00099-f006:**
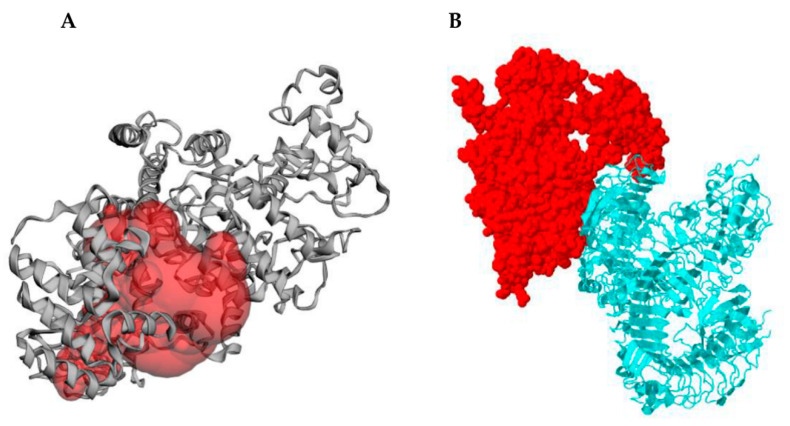
Binding pocket prediction and molecular docking of Ov-DKR-2 with TLR4. (**A**) Visualization of the largest binding pocket (in red) automatically identified on the refined 3D structure of Ov-DKR-2. This largest pocket is predicted to be between the residues 41–797, containing 138 amino acids: (**B**) Receptor (TLR-4) is presented in red whereas cyan represents the multi-epitope peptide which acts the ligand in the docked complex generated by the FRODock 2.0 server.

**Figure 7 pathogens-10-00099-f007:**
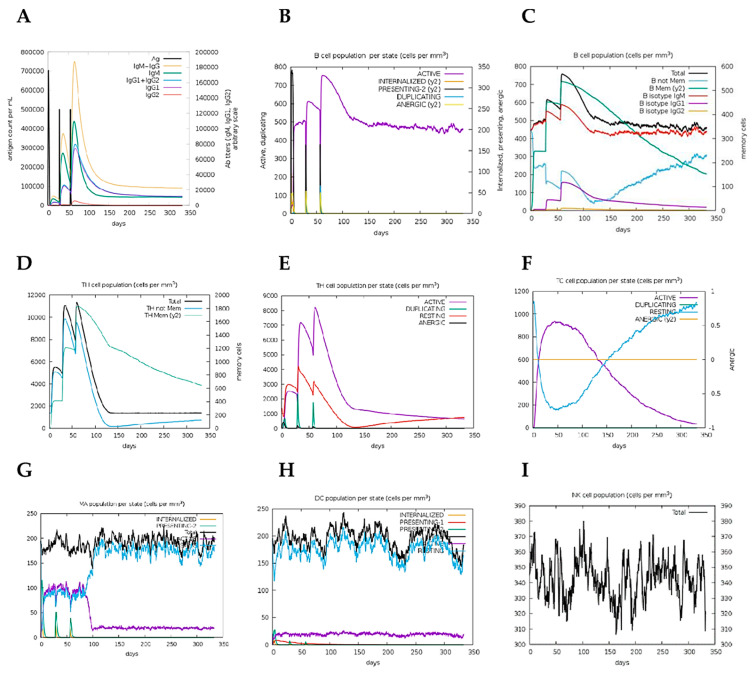
In silico simulation of immune response using the chimeric vaccine antigen, Ov-DKR-2 as antigen. Levels of antibodies, antigen and cell populations (x-axes) were plotted against time after immunization in days (y-axes) (**A**) Antigen and immunoglobulins, (**B**) B cell population, (**C**) B cell population per state, (**D**) T-helper (TH) cell population, (**E**) TH cell population per state, (**F**) T cell (TC) population per state, (**G**) macrophage (MA) population per state, (**H**) dendritic cell (DC) population per state, and (**I**) natural killer (NK) population per state.

**Figure 8 pathogens-10-00099-f008:**
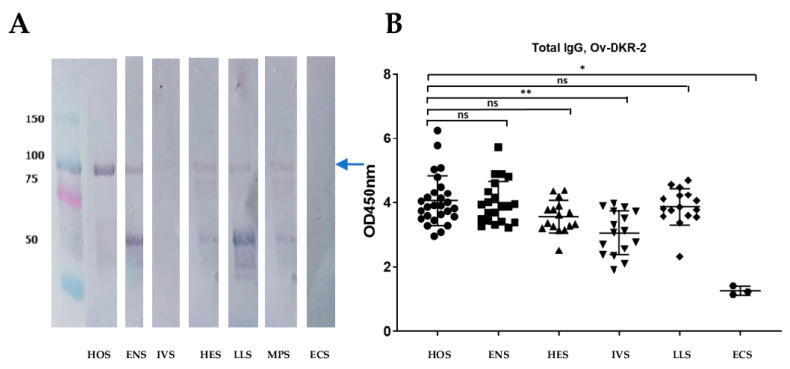
Western blot analysis and ELISA of the chimeric protein using various human serum samples. (**A**) The proteins, initially purified on a Ni^++^-IMAC column were resolved by SDS-PAGE. Proteins were transferred onto a nitrocellulose membrane and incubated with different sets of serum pools diluted at 1:800 (HOS: human onchocerciasis serum, ENS: endemic normal serum, IVS: ivermectin compliance serum, HES: hypo-endemic serum, LLS: human loiasis serum, MPS: human mansonellosis serum, ECS: European control serum. The expected size of the protein is indicated by the arrow. (**B**) Purified 8x-His tagged Ov-DKR-2 was used to coat microtiter plates. Plates were blocked with 5% milk and later incubated with serum samples from the indicated cohorts followed by secondary antibody (goat anti-human IgG peroxidase conjugate). The plates were revealed using 3,3′,5,5′-Tetramethylbenzidine (TMB) and stopped with 3 M HCl. The optical density of stopped reactions was read at 450 nm, and optical density (OD) values were plotted against serum from the different cohorts. Comparisons of more than two groups were made using the Kruskal–Wallis test (with Dunn’s or Tukey’s correction for multiple comparisons) for independent groups as appropriate. Error bars represent the standard error mean (SEM). A *p*-value < 0.05 was considered statistically significant (* *p* < 0.05 and ** *p* < 0.01). Scatter plots were generated using Graph Pad Prism 7.0 (La Jolla, CA, USA).

**Table 1 pathogens-10-00099-t001:** Conservation (percentage amino-acid identity) of the selected proteins in the indicated related nematodes, *Onchocerca ochengi, O. flexuosa, Loa loa, Brugia malayi and Wuchereria bacncrofti*.

Protein	Percentage Identity				
	*O. ochengi*	*O. flexuosa*	*L. loa*	*B. malayi*	*W. bancrofti*
Ov103	99.4	81.6	72.8	68.4	70.3
Ov-RAL-2	99.4	45.5	49.3	56.1	55.8
Ov-ASP-1	95.4	48.3	69.5	73.6	74.1
Ov-ALT-1	99.3	45.5	38.4	46.2	43.8
Ov-ALT-2	76.6	45.5	36.0	47.7	47.3
Ov-B20	97.2	Not found	89.6	85.0	88.9
Ov-RBP-1	100	81.1	80.9	80.9	80.5
Ov-CHI-1	90.0	Not found	76.1	68.6	70.3

**Table 2 pathogens-10-00099-t002:** Predicted linear B-lymphocyte epitopes and T-lymphocyte epitopes in the designed vaccine construct with their percentage conservation in related nematodes given in brackets. Serial numbers before each epitope sequence refer to the arrangement of the sequences in the final chimeric design and sequence highlighted in bold are sequences that over-lapped in the different epitope types predicted.

Protein	CTL Epitopes	HTL Epitopes	B-Epitopes
Ov-103	13. **FTDIKSLFA** (89)	13. FFTKH**FTDIKSLFA**KDEK (78)	28. NPKMDFTNKENKWEELLKKI (55)
	14. **SAPTTFATY** (78)	14. KL**KNSAPTTFATYLFTCIVPVL** (64)	29. MSMLQPLANDMQKTTLGKIG (40)
		15. TIQDKMSMLQPLANDMQK (50)	14. **KNSAPTTFATYLFTCIVPVL** (65)
Ov-RAL-2	17. FINR**LGGSY** (0)	16. PS**VIDEFYNLL**KTDENKTDQ (42)	16. RQQQQQQQQQQQRDE REIPP
	16. QTEADVEAF (22)	17. **LGGSY**KVRFTQFMEEVKKAR (4)	16. KTDENKTDQ**QTEADVEAF**IN (25)
	16. **VIDEFYNLL** (44)		
Ov-ASP-1	1. RTLIVICHY (78)	18. VLHF**TQMAWGKTYKIGC** (24)	20. SPRQQREG**VGENVYAY**WSSV (50)
	2. RSDLINGKLKNRNGTY (31)	19. CPGGKLTALERKKIVGQN NKY (39)	14. EGLKKTAGTDAGKSWWSKLP (25)
	20. G**VGENVYAY** (67)	20. **VGENVYAY**WSSVSVEGLKK (37)	18. **WGKTYKIGC**GVATQCDGGRT (85)
	6. KIVGQNNKY (44)		
	18. **TQMAWGKTY** (22)		
Ov-ALT-2	R**VDGFEHTY** (15)		3. **VDGFEHTY**CSPDENWTCQYD (20)
Ov-B20	8. ESEKQHWPY (80)	27. HSKRSFLKLEPMQQRKL (100)	
	9. HSDTRGTLI (67)	26. NIERYASLPTLHSDTRGTLIKI (81)	
	10. VTDSAQSTM (100)		
	11. RTDLSPGLW (92)		
	12. VTDASSATW (20)		
Ov-RBP-1	7. FVDEIIAKV	24. NTTKIITNEKFKRIANSFLQKN (60)	25. HQLILMALIGVIMANVVPFSMSNIPEEY (68)
Ov-CHI-1	4. DTNWSKGMY (56)	21. HDHPYVKLCPPNTFFND (53)	22. YDNEETIRMKMRWLKKEKGY (74)
	5. KSDSEQLLL (22)		23. EFAANYWLSKGMPKQKII (39)

**Table 3 pathogens-10-00099-t003:** Amino acid composition of conformation epitopes predicted in Ov-DKR-2 chimeric antigen with the number of residues in each epitope and antigenic score for the epitope as predicted on the ElliPro server.

No.	Epitope	No. of Residues	Score
1	^1^MAKLSTDELLDAFKEMTLLELSDFVKKFEETFEVTAAAPVAVAAAGAAPAGAAVE^55^, E^59^, ^63^FDVILEAAGDKKIGVIKVVREIVSGLGLKEAKDLVDGAPKPLLEKVAKEADEAKAKLEAAGATVTVK^130^	119	0.799
2	K^379^, ^381^TLGKIGGPGPGRQQQQQQQQQQQ^403^	24	0.783
3	^143^HYGGGSRSD^151^,^153^DLINGKLKNRNGTYGGGSRVDGFEHTYCSPDENWTCQYDGGGSDTNWSKGMYGGGSGTSCGEGPYGGGSKSDS^223^, H^251^	79	0.699
4	^476^ARGPGPGVL^483^, ^497^IGCGVATQCDGGRTGPGPGCPGGKLTALERKKIVGQNNKYGPGPGSPRQQREGVGENVYAYW^558^, ^562^SVEGLKKTAGTDAGKSWWSKLPGPGPGHDHPYVKLCPPNTFFNDGPGPGYDNEETIRM^619^, ^656^PGPGNTTKI^664^, ^705^FSMSNI PEEYGPGPG^719^, ^721^I	146	0.686
5	^703^VP^704^, ^758^MQQ^760^, ^762^K, ^764^GPGPGNPKMD^773^, ^775^TN^776^, ^778^ENK^780^, ^782^EE^783^, ^786^K, ^802^N, ^805^QKTTLGKIGHHHHHHHH^821^	42	0.672
6	^405^DEREIPP^412^, ^413^LE^414^	9	0579
7	^430^T, ^433^NKTDQ^437^	6	0.551
8	^321^KS^322^, ^325^AKDEKGP^331^, ^335^LK^336^, ^339^S	12	0.547
9	^645^KGMPK^649^	5	0.521
10	^291^GLWGGG^296^	6	0.512

## Data Availability

All relevant data are within the paper.

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
