# Peer review of "Computational Design and Preliminary Serological Analysis of a Novel Multi-Epitope Vaccine Candidate Against Onchocerciasis and Related Filarial Diseases"

_pathogens, 2021, doi:10.3390/pathogens10020099_

Round 1

Reviewer 1 Report

This is a very interesting manuscript and has taken what I believe to be a novel approach to vaccine development for human onchocerciasis through a predominantly computational database/ machine learning approach to identify epitopes and their use in vaccine formulation. Whilst the majority of the study is performed in silico the final assessment of cross reactivity of the resulting recombinant protein with serum from a range of infected/exposed individuals add considerable value and justification to the approaches taken, and the need to further this work.

I am not particularly familiar with the in silico approaches taken, but these seem sensible, and appear to consider both immunogenicity and vaccine safety at appropriate points in the process. Overall, the manuscript is well written and quite easy to follow, although I do have some specific points I wish to raise (below) for which I would be grateful of the authors responses.

Abstract
Line 40: I think it should be made clear here that you are referring to in silico immune stimulation.

Introduction
Line 126: "Ov-CPI-2 was eliminated because of the prevalence to its homologue in humans which may lead to autoimmunity" - I did not see any further reference to this point in either the results or materials and methods section- please can you expand on this in the relevant subsequent sections.

Figure 1: Text is a little small and difficult to read. In the legend please define CTL and HTL.

Results
Section 2.1
Line 155 (and elsewhere): Why was this particular adjuvant chosen? The need for a mixed humoral and cellular response to O. volvulus vaccines is discussed in the introduction, but I did not find any reasoning as to why L7/L12 was chosen over other potential adjuvant candidates?

Line 156: typo- I think this should refer to Figure 2?

Section 2.2, 2.3 and 2.4 (and elsewhere): What constitutes a "high-scoring"/ "high binding" and/or "top binding" epitopes? What was the scale, and your threshold for selection? This may be more important to place in the methods, but including these scores in table 1B may be useful to allow relative comparison of epitopes to one another (although presumably comparison between B- HTL and CTL epitopes would not be appropriate).

Table 1B: Similarly, is it mentioned in the methods that some candidates were both B- and T-cell epitopes. It would be useful to identify these in this table.

Section 2.7: It might be useful to explain here the multi-epitope subunit is what is subsequently referred to as Ov-DKR-2?

Figure 2: Similar comment as to Table 1B in terms of identifying which peptides were both B- and T-cell epitopes- the figure currently implies all linear B epitopes were also HTL epitopes- is that correct?

Section 2.9: If the adjuvant produced no epitopes capable of achieving the required threshold for IFN-gamma induction does this imply it is not an appropriate choice (if IFN-gamma induction is its principle purpose)? I did not see this discussed further, it would be good to have more information and interpretation of this. Similarly, the vaccine construct had a total of 668 epitopes with positive and negative prediction scores- how many had a positive prediction, and how many negative? What was the net effect? It would also be interesting to know how this would compare to a vaccine composed of the individual epitopes as separate peptide sequences if you have this data?

Figure 3: Please move the labels as they are overlaid on the plots. Also, would you consider plotting on a log scale to allow better visualization of lesser produced cytokines?

Section 2.10
Line 235: "The generated vaccine construct was predicted to be non-allergenic..." Please can you clarify if this analysis included the adjuvant or not?

Figure 4: Please clarify in the legend what colours denote what secondary structures.

Figure 7: Please can you move figure labels and enlarge the text. These plots are very hard to read currently.

Section 2.19 and Figure 8B: What were the sample sizes of each cohort? in the figure it appears the ECS group consists of three individuals- is this correct? If so, I am dubious as to the value of parametric analysis, as gaussian distribution should not be assumed with such a small sample size (I note elsewhere it is stated normality was tested for using a Shapiro-Wilks test, but again I do not think this would be informative with such a small sample size). Personally I would prefer this analysis to be repeated with an equivalent non-parametric test (eg. Kruskal-Wallis). Additionally, please define what error bars are demonstrating in figure 8B.

Discussion
Line 481: typo - toIgG

Line 501: "The E. coli expression system which is easy to use and comparatively cheaper is the preferred choice for the production of recombinant proteins". Whilst there are clear advantages to E. coli as stated, I'm not sure I agree it is the preferred choice in all cases- there are numerous examples as to why alternative systems such as yeast may be preferable, particularly when producing proteins originating from complex eukaryotic organisms.

Materials and Methods
Section 5.1 Line 563: This sentence appears to be incomplete?

Section 5.5: Why was mouse MHC-II used rather than human?

Section 5.6 Line 613: It might be useful to mention the host species for these parasites.

Section 5.7: see previous comments on "scoring" for epitopes and defining Ov-DKR-2

Section 5.18: Is dose and formulation of the vaccine administration a factor in this analysis? If so, please can you specify dosage.

Section 5.21: See previous comments about numbers of samples in each cohort.

Section 5.22: See previous comments about analysis.

Author Response

Reviewer 1: This is a very interesting manuscript and has taken what I believe to be a novel approach to vaccine development for human onchocerciasis through a predominantly computational database/ machine learning approach to identify epitopes and their use in vaccine formulation. Whilst the majority of the study is performed in silico the final assessment of cross reactivity of the resulting recombinant protein with serum from a range of infected/exposed individuals add considerable value and justification to the approaches taken, and the need to further this work.

I am not particularly familiar with the in silico approaches taken, but these seem sensible, and appear to consider both immunogenicity and vaccine safety at appropriate points in the process. Overall, the manuscript is well written and quite easy to follow, although I do have some specific points I wish to raise (below) for which I would be grateful of the authors responses.

Reviewer 1: Abstract
Line 40:
I think it should be made clear here that you are referring to in silico immune stimulation.
Authors' Response: The sentence has been corrected and the word, “in-silico” has been added at the beginning of the sentence to the communicate what is intended (line 44).

Reviewer 1: Introduction
Line 126: "Ov-CPI-2 was eliminated because of the prevalence to its homologue in humans which may lead to autoimmunity" - I did not see any further reference to this point in either the results or materials and methods section- please can you expand on this in the relevant subsequent sections.
Authors' Response: The protein, Ov-CPI-2 was eliminated from the analyses done with the 8 proteins previously reported because it has a homologue in humans which is about 29% identical – this may generate an auto-immune response. This protein was therefore not used for epitope prediction. This is the reason why no additional information is further mentioned about the protein in the results or materials and methods section.

Reviewer 1: Figure 1: Text is a little small and difficult to read. In the legend please define CTL and HTL.
Authors’ Response: The font size used for the text has been increased from 6 to 9 and the full meanings of CTL and HTL have been added to the legend for clarity purposes.

Reviewer 1: Section 2.1: Line 155 (and elsewhere): Why was this particular adjuvant chosen? The need for a mixed humoral and cellular response to O. volvulus vaccines is discussed in the introduction, but I did not find any reasoning as to why L7/L12 was chosen over other potential adjuvant candidates?
Authors’ Response: For multi-epitope vaccines, since the traditional carriers and adjuvants are associated with poor efficacy, vaccine designs with built-in adjuvants have been proposed. Therefore, a built-in adjuvant exhibiting both the functions of a transmission system and a traditional adjuvant, is constructed within the vaccine to improve the immunogenicity of epitope peptides by stimulating the innate immune response required for an adaptive immune response [1]. The Mycobacterium tuberculosis 50S ribosomal protein L7/L12 (RL7_MYCTU) P9WHE3 was retrieved from the UniProt database and used as a built-in adjuvant on the basis of the fact that it is a TLR-4 agonist and protective immunity to the larval stages of Onchocerca volvulus has been reported to be dependent on Toll-like receptor 4 [2].

Reviewer 1: Line 156: typo- I think this should refer to Figure 2?
Authors’ Response: This typographical error has been corrected and Figure 2 has been referred to (line 182).

Reviewer 1: Section 2.2, 2.3 and 2.4 (and elsewhere): What constitutes a "high-scoring"/ "high binding" and/or "top binding" epitopes? What was the scale, and your threshold for selection? This may be more important to place in the methods, but including these scores in table 1B may be useful to allow relative comparison of epitopes to one another (although presumably comparison between B- HTL and CTL epitopes would not be appropriate).
Authors’ Response: The different servers used for epitope prediction had different criteria for epitope selection. For the BCPreds server used for linear B-cell epitope prediction, epitopes selected for the final protein were epitopes that had a score of 1, while the NetMHCII 2.3 server, the epitopes selected for the construction of the chimera was based on their ability to bind to several MHCII molecules and the low IC50 scores. Generally, IC50 scores of less that 50nm indicate high affinity. On the other hand, for the NetCTL1.2 server, the default threshold for epitope selection is set at 0.75 and this default was used for epitope prediction. We agree with the reviewer that adding the scores to epitopes may be useful but we also think that the Table 1B is too saturated already and adding such information may lead to loss of legibility.

Reviewer 1: Table 1B: Similarly, is it mentioned in the methods that some candidates were both B- and T-cell epitopes. It would be useful to identify these in this table.
Author’s Response: The amino acids contained in B- and T- cells epitopes have been marked in the Table 1B. A better description has also been given in the legend on the arrangement of the epitopes in the chimera.

Reviewer 1: Section 2.7: It might be useful to explain here the multi-epitope subunit is what is subsequently referred to as Ov-DKR-2?
Authors’ Response: This has been taken into consideration and the designation of the chimeric antigen is indicated (line 232). We also just wish to indicate that the designation of the chimera as Ov-DKR-2 is made mentioned of in the abstract.

Reviewer 1: Figure 2: Similar comment as to Table 1B in terms of identifying which peptides were both B- and T-cell epitopes- the figure currently implies all linear B epitopes were also HTL epitopes- is that correct?
Authors’ Response: Not all linear B-epitopes were HTL-epitopes. They are thus arranged in Figure 2 since the fusion of these two types of epitopes was done with similar linkers. Some of the B-cell epitopes predicted had sequences that overlapped with HTL and CTL epitopes (line 193). Epitopes having overlapping sequences and as such were fused to form epitopes with contiguous amino acids. In Table 1B, the serial numbers before the epitopes indicate the sequence of arrangement in the chimera.

Reviewer 1: Section 2.9: If the adjuvant produced no epitopes capable of achieving the required threshold for IFN-gamma induction does this imply it is not an appropriate choice (if IFN-gamma induction is its principle purpose)? I did not see this discussed further, it would be good to have more information and interpretation of this. Similarly, the vaccine construct had a total of 668 epitopes with positive and negative prediction scores- how many had a positive prediction, and how many negative? What was the net effect? It would also be interesting to know how this would compare to a vaccine composed of the individual epitopes as separate peptide sequences if you have this data?
Authors’ Response: The adjuvant was selected based on the fact that it is a TLR4 agonist and TLR4 has been implicated in protective immunity against larval stages of Onchocerca volvulus. The mechanism of action of the agonist is yet to be elucidated it may therefore not be appropriate to conclude that an adjuvant is not a suitable choice on the basis of ability to stimulate IFN-gamma induction only. “This server will generate all possible overlapping peptides (of length or window selected by user) from the introduced antigen. The server will predict IFN epitopes in these overlapping peptides. This server also allows users to rank these peptides/epitopes based on their SVM score. In summary this module identifies the best antigenic regions or IFN epitope in a query antigen sequence that can induce IFN-gamma” (IFN epitope website). The total of 668 potential epitopes were just overlapping 15-mer sequences that span across the whole structure of the chimera. A total of 86 potential epitopes had positive scores and thus predicting their ability to induce IFN-gamma secretion while the 15-mers with negative scores predict the inability of these sequences to induce IFN-gamma secretion. The presence of IFN-gamma-inducing epitopes is what we were interested in; since IFN-gamma has been reported to be important in protection [3]. The negative scores on the non-inducing epitopes do not predict the ability of these sequences to abrogate or inhibit IFN-gamma secretion. Thus, it will be difficult to talk about a “net effect”.

Reviewer 1: Figure 3: Please move the labels as they are overlaid on the plots. Also, would you consider plotting on a log scale to allow better visualization of lesser produced cytokines?
Authors’ Response: We agree with the reviewer on the need to move the labels as well as also performing a plot on a log scale. However, the C-ImmSim server that was used for immune simulation generates the results automatically and gives no interface for the user to decide how to label graphs and choose scales.

Reviewer 1: Section 2.10 Line 235: "The generated vaccine construct was predicted to be non-allergenic..." Please can you clarify if this analysis included the adjuvant or not?.
Authors’ Response: The statement above refers to the whole sequence of the chimera, including the various epitopes and the built-in adjuvant.

Reviewer 1: Figure 4: Please clarify in the legend which colours denote which secondary structures.
Authors' Response: This comment has been taken into consideration and colours denoting the different secondary structures have been added in the legend (line 300-301).

Reviewer 1: Figure 7: Please can you move figure labels and enlarge the text. These plots are very hard to read currently.
Authors’ Response: We understand the reviewer's concerns and agree that there is a need to improve on the figure but we wish to note that the plots in Figure 7 are auto-generated by the web-based C-ImmSim tool. The users have no control over the figure labelling. We also noticed that zooming on the figure helps increase legibility.

Reviewer 1: Section 2.19 and Figure 8B: What were the sample sizes of each cohort? in the figure it appears the ECS group consists of three individuals- is this correct? If so, I am dubious as to the value of parametric analysis, as gaussian distribution should not be assumed with such a small sample size (I note elsewhere it is stated normality was tested for using a Shapiro-Wilks test, but again I do not think this would be informative with such a small sample size). Personally, I would prefer this analysis to be repeated with an equivalent non-parametric test (eg. Kruskal-Wallis). Additionally, please define what error bars are demonstrating in figure 8B.
Authors’ Response: For cohorts used, the numbers were as follows: HOS (27), ENS (21), HES (16), IVS (16), LLS (16) and ECS (3). We agree with the reviewer that it will be difficult to assume Gaussian distribution for a small sample size like the ECS cohort. We have therefore repeated the analyses with the Kruskal-Wallis test as recommended and the error bars have also been defined.

Reviewer 1: Discussion: Line 481 typo: toIgG
Authors’ Response: This typographical error has been corrected (line 575).

Reviewer 1: Line 501: "The E. coli expression system which is easy to use and comparatively cheaper is the preferred choice for the production of recombinant proteins". Whilst there are clear advantages to E. coli as stated, I'm not sure I agree it is the preferred choice in all cases- there are numerous examples as to why alternative systems such as yeast may be preferable, particularly when producing proteins originating from complex eukaryotic organisms.
Authors’ Response: The above statement has been corrected to present the proper reality and context (line 603-605).

Reviewer 1: Materials and Methods: Section 5.1 Line 563: This sentence appears to be incomplete?
Authors’ Response: We agree and the sentenced has been completed to read, “…and the serum samples are described in our previous publication” (line 670).

Reviewer 1: Section 5.5: Why was mouse MHC-II used rather than human?
Authors’ Response: The NetMHCII 2.3 Server (http://www.cbs.dtu.dk/services/NetMHCII/) [115] was used to predict HTL epitopes of 15-mer length for human alleles. NetMHCII 2.2 server predicts binding of peptides to HLA-DR, HLA-DQ, HLA-DP and mouse MHC class II alleles using artificial neuron networks (602-605). The server does have the ability to predict epitopes that can bind to both mouse and human alleles but only epitopes that bind to human alleles were predicted in the present work.

Reviewer 1: Section 5.6 Line 613: It might be useful to mention the host species for these parasites.
Authors' Response: This comment has been taken into consideration on the hosts species for the various parasites have been included (732-736).

Reviewer 1: Section 5.7: See previous comments on "scoring" for epitopes and defining Ov-DKR-2
Authors’ Response: This concern has been addressed.

Reviewer 1: Section 5.18: Is dose and formulation of the vaccine administration a factor in this analysis? If so, please can you specify dosage.
Authors’ Response: Dosage and formulation were not factors considered in these simulations. A constant number of 1000 antigens were given per injection and apart from the epitopes, built-in adjuvant and linkers used to construct the chimera, no other additives or adjuvants were included in the vaccine formulation. It is important to indicate, however, that the C-ImmSim server accepts only protein sequences and it is therefore difficult to make predictions on how the different adjuvants will stimulate the immune system.

Reviewer 1: Section 5.21: See previous comments about numbers of samples in each cohort.
Authors’ Response: This concern has been addressed.

Reviewer 1: Section 5.22: See previous comments about analysis.
Authors’ response: This concern has been addressed.

References

  1. Lei, Y., et al., Application of built-in adjuvants for epitope-based vaccines. PeerJ, 2019. 6: p. e6185-e6185.
  2. Kerepesi, L.A., et al., Protective immunity to the larval stages of onchocerca volvulus is dependent on Toll-like receptor 4. Infect Immun, 2005. 73(12): p. 8291-7.
  3. Elson, L.H., et al., Immunity to onchocerciasis: putative immune persons produce a Th1-like response to Onchocerca volvulus. J Infect Dis, 1995. 171(3): p. 652-8.

Reviewer 2 Report

Paper is addressing a topic that so far received insufficient international attention: onchocerciasis vaccination.   

We agree with the authors that further research in this domain is needed. The interest of this paper is that it hopefully will stimulate research of other researchers in this direction.

The paper is well written and very well documented.

The same research group recently published a paper in pathogens “In Silico Design and Validation of OvMANE1, a Chimeric Antigen for Human Onchocerciasis Diagnosis” This reference is cited when the authors mention doxycycline resistance of Wolbachia? Not clear to me.

Minor Comment
In the abstract it is mentioned “The elimination plan for the disease is currently challenged by many factors including amongst others resistance to the main chemotherapeutic agent, ivermectin (IVM)”. Ivermectin resistance is later also mentioned in the introduction.

I’m not sure ivermectin resistance has been demonstrated in onchocerciasis. Indeed IVM always kills the mf very rapidly but its effect on the fertility of the female worm may decrease. It is probably better to use the term “sub-optimal treatment”.

In the introduction the authors state “though infections were previously associated mainly with skin and eye lesions, recent records show a trend towards increasing mortality. Results from data collected over a 27-year period reported a 5.9% mortality risk attributable to onchocerciasis (3) with blindness in adults giving rise to a significant increase in mortality and reduced life expectancy (4).”

I propose to propose to separate the statement about the study by Walker et al (ref 3) and Pion et al (ref 4)

Indeed the study by Walker et al needs some additional explanation. The excess mortality reported by Walker et al most likely was not caused by blindness. Indeed in the study by Walker et al the relative risk of mortality, for a given density, was statistically significantly higher in individuals younger than 20 years than in those aged above 20. A more likely hypothesis is that this excess of mortality under 20 years was a consequence of onchocerciasis-associated epilepsy, a condition affecting children and adolescents while river blindness generally occurs after the age of 20.

The hypothesis above was formulated in the following paper

Colebunders R, Siewe Fodjo JN, Hopkins A, Hotterbeekx A, Lakwo TL, Kalinga A, Logora MY, Basáñez MG. From river blindness to river epilepsy: Implications for onchocerciasis elimination programmes. PLoS Negl Trop Dis. 2019 Jul 18;13(7):e0007407

Blindness is also associated with reduced life expectancy. So the statement about the study of Pion et al is OK

Author Response

Reviewer 2

Comments and Suggestions for Authors

Reviewer 2: Paper is addressing a topic that so far received insufficient international attention: onchocerciasis vaccination.   

We agree with the authors that further research in this domain is needed. The interest of this paper is that it hopefully will stimulate research of other researchers in this direction.

The paper is well written and very well documented.

The same research group recently published a paper in pathogens “In Silico Design and Validation of OvMANE1, a Chimeric Antigen for Human Onchocerciasis Diagnosis” This reference is cited when the authors mention doxycycline resistance of Wolbachia? Not clear to me.
Authors’ Response: We acknowledge this error and the right reference has been inserted for the sentence (line 83).

Reviewer 2: Minor Comment
In the abstract it is mentioned “The elimination plan for the disease is currently challenged by many factors including amongst others resistance to the main chemotherapeutic agent, ivermectin (IVM)”. Ivermectin resistance is later also mentioned in the introduction.

I’m not sure ivermectin resistance has been demonstrated in onchocerciasis. Indeed IVM always kills the mf very rapidly but its effect on the fertility of the female worm may decrease. It is probably better to use the term “sub-optimal treatment”.
Authors’ Response: This comment has been taken into consideration and “resistance to IVM” has been modified throughout the manuscript to reflect what is recommended by the reviewer.

Reviewer 2: In the introduction the authors state “though infections were previously associated mainly with skin and eye lesions, recent records show a trend towards increasing mortality. Results from data collected over a 27-year period reported a 5.9% mortality risk attributable to onchocerciasis (3) with blindness in adults giving rise to a significant increase in mortality and reduced life expectancy (4).”

I propose to propose to separate the statement about the study by Walker et al (ref 3) and Pion et al (ref 4)

Indeed the study by Walker et al needs some additional explanation. The excess mortality reported by Walker et al most likely was not caused by blindness. Indeed in the study by Walker et al the relative risk of mortality, for a given density, was statistically significantly higher in individuals younger than 20 years than in those aged above 20. A more likely hypothesis is that this excess of mortality under 20 years was a consequence of onchocerciasis-associated epilepsy, a condition affecting children and adolescents while river blindness generally occurs after the age of 20.

The hypothesis above was formulated in the following paper

Colebunders R, Siewe Fodjo JN, Hopkins A, Hotterbeekx A, Lakwo TL, Kalinga A, Logora MY, Basáñez MG. From river blindness to river epilepsy: Implications for onchocerciasis elimination programmes. PLoS Negl Trop Dis. 2019 Jul 18;13(7):e0007407

Blindness is also associated with reduced life expectancy. So the statement about the study of Pion et al is OK.
Authors’ Response: The statement with the two references has been separated and explained to provide the context and reality (line 61-64).

Round 2

Reviewer 1 Report

The authors have responded to my comments and queries and I feel they have explained these in detail. I am, however, somewhat confused as to why they responded in such detail as responses as opposed to making changes in the manuscript itself? The reasons for my queries in the first place were because the information was not provided in the manuscript. In several cases, whilst they have adequately explained their rationale and provided extra detail as responses, much of this useful information has not made its way into the manuscript itself- as a consequence I can envisage other readers raising the same questions without being provided with the answers the authors have already put so much time and effort into in this review process. I would therefore encourage the authors review and insert some of these responses into the manuscript itself where relevant (mostly in the methods section) including the additional information, data and references provided in response to my queries.

Author Response

Dear Reviewer,

We are so grateful for your comments, suggestions and time to our manuscript. We agree that it was important to add the explanations to the manuscript and this was done accordingly as you could see marked in yellow within the submitted revised version. Point-by-point, the lines where the new information is included in the manuscript are as follows, indicated in red:

Reviewer 1: Line 40: I think it should be made clear here that you are referring to in silico immune stimulation.
Authors' Response: The sentence has been corrected and the word, “in-silico” has been added at the beginning of the sentence to the communicate what is intended (line 44).

Reviewer 1: Introduction: Line 126: "Ov-CPI-2 was eliminated because of the prevalence to its homologue in humans which may lead to autoimmunity" - I did not see any further reference to this point in either the results or materials and methods section- please can you expand on this in the relevant subsequent sections.
Authors' Response: The protein, Ov-CPI-2 was eliminated from the analyses done with the 8 proteins previously reported because it has a homologue in humans which is about 29% identical – this may generate an auto-immune response. This protein was therefore not used for epitope prediction. More information on Ov-CPI-2 has been added to the manuscript (147-149, 174, 513-516).

Reviewer 1: Section 2.1: Line 155 (and elsewhere): Why was this particular adjuvant chosen? The need for a mixed humoral and cellular response to O. volvulus vaccines is discussed in the introduction, but I did not find any reasoning as to why L7/L12 was chosen over other potential adjuvant candidates?
Authors' Response: For multi-epitope vaccines, since the traditional carriers and adjuvants are associated with poor efficacy, vaccine designs with built-in adjuvants have been proposed. Therefore, a built-in adjuvant exhibiting both the functions of a transmission system and a traditional adjuvant, is constructed within the vaccine to improve the immunogenicity of epitope peptides by stimulating the innate immune response required for an adaptive immune response [1]. The Mycobacterium tuberculosis 50S ribosomal protein L7/L12 (RL7_MYCTU) P9WHE3 was retrieved from the UniProt database and used as a built-in adjuvant on the basis of the fact that it is a TLR-4 agonist and protective immunity to the larval stages of Onchocerca volvulus has been reported to be dependent on Toll-like receptor 4 [2]. This information has been added to the manuscript (183-191).

Reviewer 1: Line 156: typo- I think this should refer to Figure 2?
Authors' Response: This typographical error has been corrected and Figure 2 has been referred to (line 191).

Reviewer 1: Section 2.2, 2.3 and 2.4 (and elsewhere): What constitutes a "high-scoring"/ "high binding" and/or "top binding" epitopes? What was the scale, and your threshold for selection? This may be more important to place in the methods, but including these scores in table 1B may be useful to allow relative comparison of epitopes to one another (although presumably comparison between B- HTL and CTL epitopes would not be appropriate).
Authors' Response: The different servers used for epitope prediction had different criteria for epitope selection. For the BCPreds server used for linear B-cell epitope prediction, epitopes selected for the final protein were epitopes that had a score of 1 (722-723), while the NetMHCII 2.3 server, the epitopes selected for the construction of the chimera was based on their ability to bind to several MHCII molecules and the low IC50 scores (750-751). Generally, IC50 scores of less that 50nm indicate high affinity. On the other hand, for the NetCTL1.2 server, the default threshold for epitope selection is set at 0.75 and this default was used for epitope prediction (736-737). We agree with the reviewer that adding the scores to epitopes may be useful but we also think that the Table 1B is too saturated already and adding such information may lead to loss of legibility. Information on the criteria for epitope selection has also been added to the manuscript and line numbers are indicated above.

Reviewer 1: Table 1B: Similarly, is it mentioned in the methods that some candidates were both B- and T-cell epitopes. It would be useful to identify these in this table.
Authors' Response: The amino acids contained in B- and T- cells epitopes have been marked in the Table 1B. A better description has also been added in the legend on the arrangement of the epitopes in the chimera (lines 230-231).

Reviewer 1: Section 2.7: It might be useful to explain here the multi-epitope subunit is what is subsequently referred to as Ov-DKR-2?
Authors' Response: This has been taken into consideration and the designation of the chimeric antigen is indicated (line 245). We also just wish to indicate that the designation of the chimera as Ov-DKR-2 is made mentioned of in the abstract.

Reviewer 1: Figure 2: Similar comment as to Table 1B in terms of identifying which peptides were both B- and T-cell epitopes- the figure currently implies all linear B epitopes were also HTL epitopes - is that correct?
Authors' Response: Not all linear B-epitopes were HTL-epitopes. They are thus arranged in Figure 2 since the fusion of these two types of epitopes was done with similar linkers. Some of the B-cell epitopes predicted had sequences that overlapped with HTL and CTL epitopes. Epitopes having overlapping sequences and as such were fused to form epitopes with contiguous amino acids (lines 205-207). In Table 1B, the serial numbers before the epitopes indicate the sequence of arrangement in the chimera.

Reviewer 1: Section 2.9: If the adjuvant produced no epitopes capable of achieving the required threshold for IFN-gamma induction does this imply it is not an appropriate choice (if IFN-gamma induction is its principle purpose)? I did not see this discussed further, it would be good to have more information and interpretation of this. Similarly, the vaccine construct had a total of 668 epitopes with positive and negative prediction scores- how many had a positive prediction, and how many negative? What was the net effect? It would also be interesting to know how this would compare to a vaccine composed of the individual epitopes as separate peptide sequences if you have this data?
Authors' Response: The adjuvant was selected based on the fact that it is a TLR4 agonist and TLR4 has been implicated in protective immunity against larval stages of Onchocerca volvulus. The mechanism of action of the agonist is yet to be elucidated it may therefore not be appropriate to conclude that an adjuvant is not a suitable choice on the basis of ability to stimulate IFN-gamma induction only. “This server will generate all possible overlapping peptides (of length or window selected by user) from the introduced antigen. The server will predict IFN epitopes in these overlapping peptides. This server also allows users to rank these peptides/epitopes based on their SVM score. In summary this module identifies the best antigenic regions or IFN epitope in a query antigen sequence that can induce IFN-gamma” (IFNepitope website). The total of 668 potential epitopes were just overlapping 15-mer sequences that span across the whole structure of the chimera. A total of 86 potential epitopes had positive scores and thus predicting their ability to induce IFN-gamma secretion while the 15-mers with negative scores predict the inability of these sequences to induce IFN-gamma secretion. The presence of IFN-gamma-inducing epitopes is what we were interested in; since IFN-gamma has been reported to be important in protection [3]. The negative scores on the non-inducing epitopes do not predict the ability of these sequences to abrogate or inhibit IFN-gamma secretion. Thus, it will be difficult to talk about a “net effect”. This explanation has been added to the manuscript (600-607).

Reviewer 1: Section 2.10: Line 235: "The generated vaccine construct was predicted to be non-allergenic..." Please can you clarify if this analysis included the adjuvant or not?.
Authors' Response: The statement above refers to the sequence of the chimera, including the various epitopes and linkers with/without the built-in adjuvant. This statement has been corrected in the manuscript (lines 286-287).

Reviewer 1: Figure 4: Please clarify in the legend what colours denote what secondary structures.
Authors' Response: This comment has been taken into consideration and colours denoting the different secondary structures have been added in the legend (line 311-312).

Reviewer 1: Section 2.19 and Figure 8B: What were the sample sizes of each cohort? in the figure it appears the ECS group consists of three individuals- is this correct? If so, I am dubious as to the value of parametric analysis, as gaussian distribution should not be assumed with such a small sample size (I note elsewhere it is stated normality was tested for using a Shapiro-Wilks test, but again I do not think this would be informative with such a small sample size). Personally, I would prefer this analysis to be repeated with an equivalent non-parametric test (eg. Kruskal-Wallis). Additionally, please define what error bars are demonstrating in figure 8B.
Authors' Response: For cohorts used, the numbers were as follows: HOS (27), ENS (21), HES (16), IVS (16), LLS (16) and ECS (3). This information has been provided in the manuscript (1052-1053). We agree with the reviewer that it will be difficult to assume Gaussian distribution for a small sample size like the ECS cohort. We have therefore repeated the analyses with the Kruskal-Wallis test as recommended and the error bars have also been defined (lines 1069-1070,435-437).

Reviewer 1: Line 501: "The E. coli expression system which is easy to use and comparatively cheaper is the preferred choice for the production of recombinant proteins". Whilst there are clear advantages to E. coli as stated, I'm not sure I agree it is the preferred choice in all cases- there are numerous examples as to why alternative systems such as yeast may be preferable, particularly when producing proteins originating from complex eukaryotic organisms.
Authors' Response: The above statement has been corrected to present the proper reality and context. The statement now reads, “The Escherichia coli expression system which is easy to use and comparatively cheaper is the preferred choice and common first step for the production of most recombinant proteins” (line 629-631).

Reviewer 1: Materials and Methods: Section 5.1 Line 563: This sentence appears to be incomplete?
Authors' Response: We agree and the sentenced has been completed to read, “…and the serum samples are described in our previous publication” (line 696).

Reviewer 1: Section 5.5: Why was mouse MHC-II used rather than human?
Authors' Response: The NetMHCII 2.3 Server (http://www.cbs.dtu.dk/services/NetMHCII/) was used to predict HTL epitopes of 15-mer length for human alleles. NetMHCII 2.3 server predicts binding of peptides to human HLA-DR, HLA-DQ, HLA-DP and mouse MHC class II alleles using artificial neuron networks (746-748). The server does have the ability to predict epitopes that can bind to both mouse and human alleles but only epitopes that bind to human alleles were predicted in the present work (line 747).

Reviewer 1: Section 5.6: Line 613: It might be useful to mention the host species for these parasites.
Authors' Response: This comment has been taken into consideration on the hosts species for the various parasites have been included (lines 760-765).

Reviewer 1: Section 5.18: Is dose and formulation of the vaccine administration a factor in this analysis? If so, please can you specify dosage.
Authors' Response: Dosage and formulation were not factors considered in these simulations. A constant number of 1000 antigens were given per injection and apart from the epitopes, built-in adjuvant and linkers used to construct the chimera, no other additives or adjuvants were included in the vaccine formulation. It is important to indicate, however, that the C-ImmSim server accepts only protein sequences and it is therefore difficult to make predictions on how the different adjuvants will stimulate the immune system. This information has been added to the manuscript (lines 585-587, 622-624).

Reviewer 2 Report

The authors correctly responded to my comments and adapted their text accordingly

Author Response

Dear Reviewer,

We are so grateful for your comments, suggestions and time to our manuscript. Your approval of our responses to your comments is highly appreciated.

With Kindest Regards